# Detecting horizontal gene transfer among microbiota: an innovative pipeline for identifying co-shared genes within the mobilome through advanced comparative analysis

Jana Schwarzerova,[1,2] Michal Zeman,[3] Vladimir Babak,[3] Katerina Jureckova,[1] Marketa Nykrynova,[1] Margaret Varga,[4] Wolfram Weckwerth,[2,5] Monika Dolejska,[6,7,8,9] Valentine Provaznik,[1,10] Ivan Rychlik,[3] Darina Cejkova[1]

**ABSTRACT**  The study presents an innovative pipeline for detecting horizontal gene transfer (HGT) among a collection of sequenced genomes from gut microbiota. Herein, chicken and porcine gut microbiota were analyzed. Based on statistical analysis, we propose that nearly identical genes co-shared between distinct genera can be evidence for a previous event of mobilization of that gene from genome to genome *via* HGT. Data mining, computational analysis, and network analysis were used to investigate genomes of 452 isolates of chicken or porcine origin to detect genes involved in HGT. The proposed pipeline is user-friendly and includes network visualization. The study highlights that different species and strains of the same genera typically carry different cargo of mobilized genes. The pipeline is capable of identifying not yet characterized genes, as well as genes that are usually co-transferred with genes involved in resistance, virulence, and/or mobilization. Among the analyzed genome collection, the main reservoirs of the HGT genes were found in *Phocaeicola* spp. (*Bacteroidaceae*) and UBA9475 spp. (early *Pseudoflavonifractor*, *Oscillospiraceae*). Altogether, over 6,000 genes suspected of HGT were identified. Genes associated with intracellular trafficking and secretion and DNA repair were enriched, while genes of unknown and general functions were dominant but not enriched. Only 15 genes were co-shared between Gram-positive and Gram-negative bacteria, mostly genes directly associated with mobilome or antibiotic resistance. However, most HGTs were identified among different genera of the same phylum. Therefore, we suggest that a significant selection pressure exists on gene variants at the phylum level.

**IMPORTANCE**  Horizontal gene transfer (HGT) is a key driver in the evolution of bacterial genomes. The acquisition of genes mediated by HGT may enable bacteria to adapt to ever-changing environmental conditions. Long-term application of antibiotics in intensive agriculture is associated with the dissemination of antibiotic resistance genes among bacteria with the consequences causing public health concern. Commensal farm-animal-associated gut microbiota are considered the reservoir of the resistance genes. Therefore, in this study, we identified known and not-yet characterized mobilized genes originating from chicken and porcine fecal samples using our innovative pipeline followed by network analysis to provide appropriate visualization to support proper interpretation.

**KEYWORDS**  animal microbiome, genome evolution, mobile genetic elements, mobilome, resistance genes, horizontal gene transfer, gut microbiota

Horizontal gene transfer (HGT) or lateral gene transfer is defined as the exchange of genetic information between organisms that are not in a parent-offspring

**Ad Hoc Peer Reviewers** Johannes Wöstemeyer, Friedrich Schiller University Jena, Jena, Germany; Shay Tal, Israel Oceanographic and Limnological Research Institute, Eilat, Israel; Alejandro Piña-Iturbe, Pontificia Universidad Catolica de Chile, Santiago, Chile; Andrew S. Lang, Memorial University of Newfoundland, St. John's, Newfoundland and Labrador, Canada

Address correspondence to Darina Cejkova, cejkovad@vut.cz.

The authors declare no conflict of interest.

See the funding table on p. 20.

relationship (1). HGT can involve the transfer of DNA between mitochondrial, nuclear, and chloroplast genomes, between exons or introns, or even among different bacteria and archaea occupying the same niche. In prokaryotes, this is one of the major drivers of genome evolution next to recombination events. HGT-mediated gene gain and loss is often the consequence of adaptation to environmental changes under strong selection pressure such as in the case of multidrug-resistant bacteria (2). On the other hand, HGT can also be a cause of adaptation: by acquiring genes from other organisms, microorganisms can rapidly gain new functions, traits, or metabolic pathways that improve their ability to compete with other microorganisms occurring in the same environment (3, 4).

Recent technological advances have opened extensive options for the cultivation of bacteria and sequencing of large amounts of data for as yet undescribed prokaryotes. However, as part of the search for the same genes across different bacteria and the correct detection of the movement of mobile gene elements (MGEs) (5–7), it is completely important to unify annotation tools (8) with a possible control of the visualization and thus create a gold standard procedure for consistent and reproducible results (9). The commonly used inference annotation approaches are based on orthologous analysis and rely on two main categories (10): graph-based and tree-based methods. The conventional methods of HGT detection, particularly in clinical settings, relied on the comparative genomic analysis of closely related taxa (11) or the analysis of complete genome sequences of multidrug-resistant pathogens, especially of *Enterobacteriaceae* family (12). It revealed that many resistance genes have not evolved within the sequenced strains but were obtained by HGT (13).

The animal gut represents one of the most dynamic environments for commensal and pathogenic bacteria. Massive antibiotic usage in farming in the past led to the selection pressure on commensal bacteria to adapt to these changes by the acquisition of antibiotic resistance genes (ARG) *via* HGT (14–16). Just like studies focused on human-associated microbiota (17), research into the composition of microbiomes in animal guts, along with the dynamics and rearrangements of their genomes, including HGT events, provides valuable insights into microbial evolution, functional capabilities, interactions, and their impacts. These insights extend beyond animal health, influencing ecosystems and even human health. Since most intestinal tract bacteria are strictly anaerobic, the usage of culture-independent techniques (such as metagenomics sequencing) represents a powerful tool for the in-depth characterization of resistome and mobilome of animal origin. However, the approach has its limitations (13) mainly due to the challenge of accurately assembling genomes for each individual. Although several recent technological advances could help improve metagenome assemblies (18), culture-dependent techniques followed by whole-genome sequencing can address the questions regarding the prevalence of MGEs among animal gut microbiota, individual bacterial capacity to harbor MGEs, and understanding the role of hypothetical genes in HGTs.

Numerous bioinformatics tools have been developed to detect HGTs within sequenced bacterial genomes. These tools can be mainly categorized into composition-based approaches, phylogenetic approaches, "best-match" methods, or combinations of various tools (19). Many of these tools are also applicable in metagenomics studies (20). Composition-based approaches analyze specific features of DNA sequences, such as nucleotide composition (GC content, oligonucleotide frequencies) or codon usage; examples include AlienHunter (21), ShadowCaster (19). Implicit phylogenetic approaches analyze patterns in sequence data without directly constructing phylogenetic trees. These methods assess various aspects of the sequences themselves, including sequence similarity, genetic distance, or the presence of shared motifs. Examples represent HGTector (22) and ShadowCaster (19). Explicit phylogenetic approaches in the detection of HGT involve constructing and analyzing phylogenetic trees to identify incongruences or anomalies that might indicate the presence of horizontally transferred genes, as in MetaCHIP (23). The "best-match" or similarity approach entails comparing a gene

sequence from one organism to a database of sequences from other organisms and identifying the sequence with the highest similarity or "best match." If the best match is from a different species or lineage than the query gene, it may suggest the possibility of HGT. The method is applied in HGTector and MetaCHIP. In addition, mathematical and probabilistic approaches also have been developed, for example, gene synteny analysis identifying cases where genes are situated in orders or positions that deviate from the expected arrangement. Such deviations are often indicative of HGT (24).

In this study, we propose an innovative *in silico* approach to detect HGT between genome sequences of cultivated bacteria. This approach combines a similarity-based approach with the phylogenetic analysis of the data set; the proposed pipeline searches for nearly identical genes co-shared by different genera, genes likely suspicious for the HGT. Recently, we initiated a systemic culture of chicken and porcine gut anaerobes subsequently followed by whole-genome sequencing and analysis to collect commensal bacteria. Using this approach, we detected known mobile genes, but more importantly, we also identified genes of unknown function to be mobilized by MGEs.

## RESULTS AND DISCUSSION

### Bacterial diversity of animal gut microbiota

Genomes of 452 bacterial isolates from animal gut microbiota were analyzed in this study (see Materials and Methods) since densely inhabited chicken and porcine gut microbiota are considered to be the source of MGEs, especially MGEs associated with the transfer and dissemination of antibiotic resistance genes (ARGs) *via* animal waste to the environment (25). Altogether, the collection of gut pure cultures comprises eight phyla: *Firmicutes* (245 isolates), *Bacteroidetes* (113 isolates), *Actinobacteria* (65 isolates), *Proteobacteria* (19 isolates), *Fusobacteria* (seven isolates), *Verrucomicrobia* (one isolate), *Elusimicrobia* (one isolate), and *Synergistetes* (one isolate) and spanning across 35 bacterial families. The phylogenetic analysis using sequences of 81 conserved bacterial genes clearly indicates the monophyletic origin of *Bacteroidetes*, *Actinobacteria,* and *Proteobacteria* phyla. On the contrary, *Firmicutes* phylum represents a polyphyletic group composed of four different phylogenetic units. The placement of the *Fusobacteriaceae* is debatable, especially if we also consider phylogenetic relatedness based on the 16S rDNA gene (Fig. 1; Fig. S1).

Similar findings were also observed by Parks et al. (26) during the construction and analysis of the Genome Taxonomy Database (GTDB), a database used for novel phylogeny-based bacterial taxonomy (27). A minor incongruence between phylogenetic analysis based on UBCG and 16S rDNA genes was found in the cluster composed of *Enterococcaceae* and *Streptococcaceae*; the bifurcation into two separate families has been found only in the UBCG tree but not 16S rDNA tree (Fig. 1; Fig. S1).

### Identification of putative HGT using pipeline

To identify and characterize HGT genes, we developed an approach based on the detection of nearly identical genes shared by different bacterial members. The experimental pipeline has been divided into three sections (Fig. 2): (i) Annotation: all examined genomes were annotated by PROKKA and protein-coding sequences ≥300 bp were extracted; (ii) Phylogenetic framework: individual genomic species (Genomospecies), genera (Genera_16S), families (Families_16S) were identified using up-to-date bacterial core genome (UBCG), Fig. 1 and 16S rDNA phylogenetic trees, Fig. S1, and comparative genomics tool dREP. In addition, conventional microbiological families and phyla were also retained. For every operational taxonomic unit (e.g., individual genomospecies and genus), the non-redundant pan-pangenome was constructed using extracted protein-coding sequences; (iii) Non-redundant pan-genomospecies analysis: nearly identical genes shared by ≥2 operational taxonomic units were identified using CD-HIT and functionally characterized by eggNOG-mapper. Then, a group of genes likely representing true genes associated with HGT was determined based on statistics analysis (Dunns' test), and the HGT genes were analyzed and visualized *via* networks.

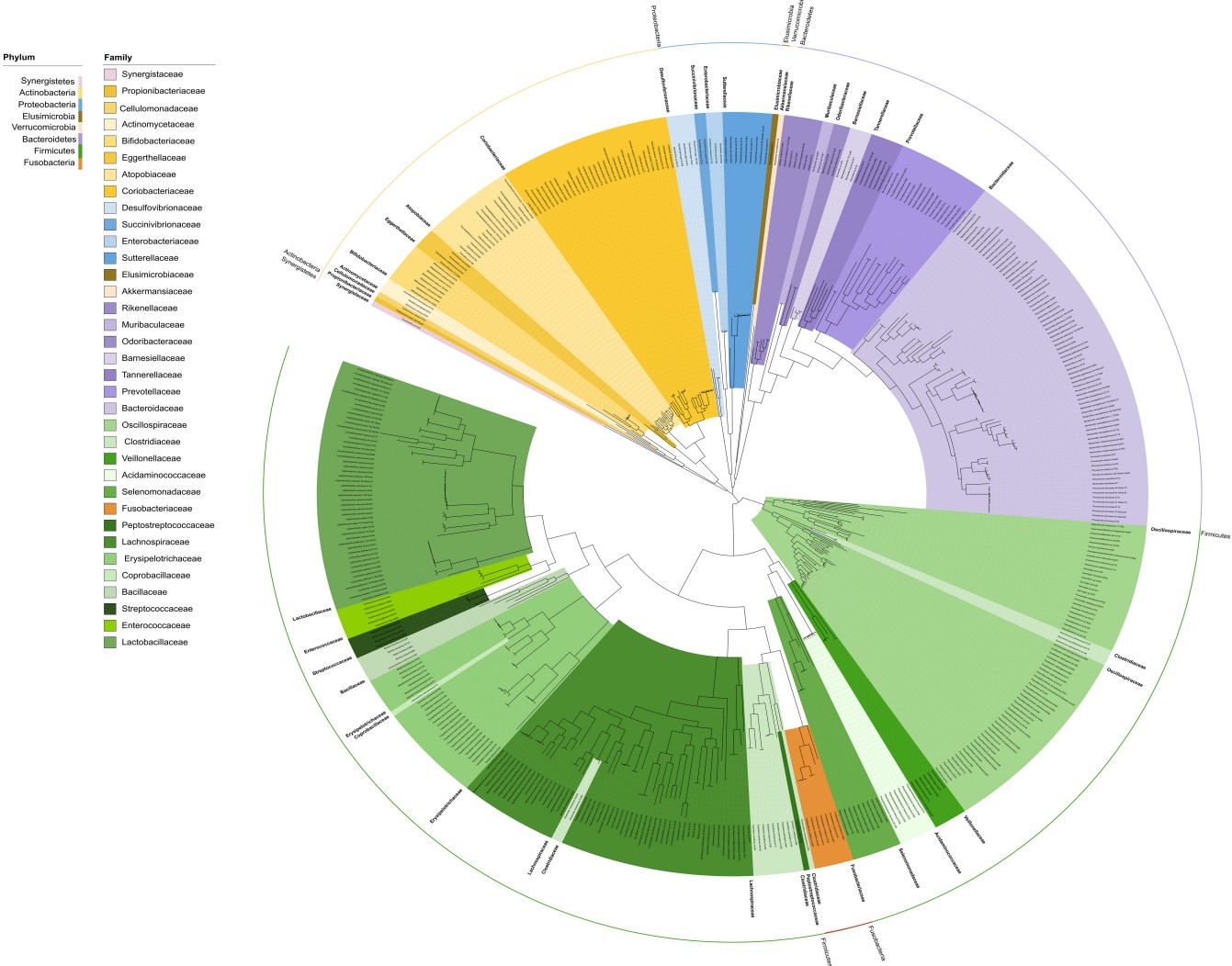

**FIG 1** Core genome phylogenetic tree inferred by the UBCG pipeline using 81 bacterial core gene sequences. Families within the phylum *Bacteroidetes* are depicted in shades of purple, within the phylum *Firmicutes* are in green, within the phylum *Proteobacteria* are in blue, within the phylum *Actinobacteria* are in yellow, within the phylum *Fusobacteria* in orange, within the phylum *Verrucomicrobia* are in beige, within the phylum *Elusimicrobia* are in golden brown, and within the phylum *Synergistetes* are in pink.

In the pipeline, we focused on the identification of nearly identical genes ≥300 bp, co-shared genes with ≥99% nucleotide identity over ≥99% global length alignment. The 300 bp cutoff was used based on the literature search, and the aim to identify real protein-coding genes, especially in the case of hypothetical genes. Several authors have employed a 500 bp threshold for their analyses (28–30), while others have used a lower threshold (200 bp) (23) to include the detection of shorter sequences, such as insertion sequence (IS) elements and recombination directionality factors. IS elements frequently constitute components of composite transposons within MGEs, which are regions challenging to accurately assemble using short-read sequencing techniques. On the other hand, we anticipated the identification of numerous hypothetical genes. To prevent an excessive representation of short hypothetical genes, we moderately raised the threshold to 300 bp.

More stringent conditions (100% nucleotide identity) are often used to detect genes that have been very recently transferred horizontally in a particular niche (28, 29). We used less stringent conditions because our collection of bacterial isolates has been collected for over 5 years and comprises isolates originating from chicken and pigs. To

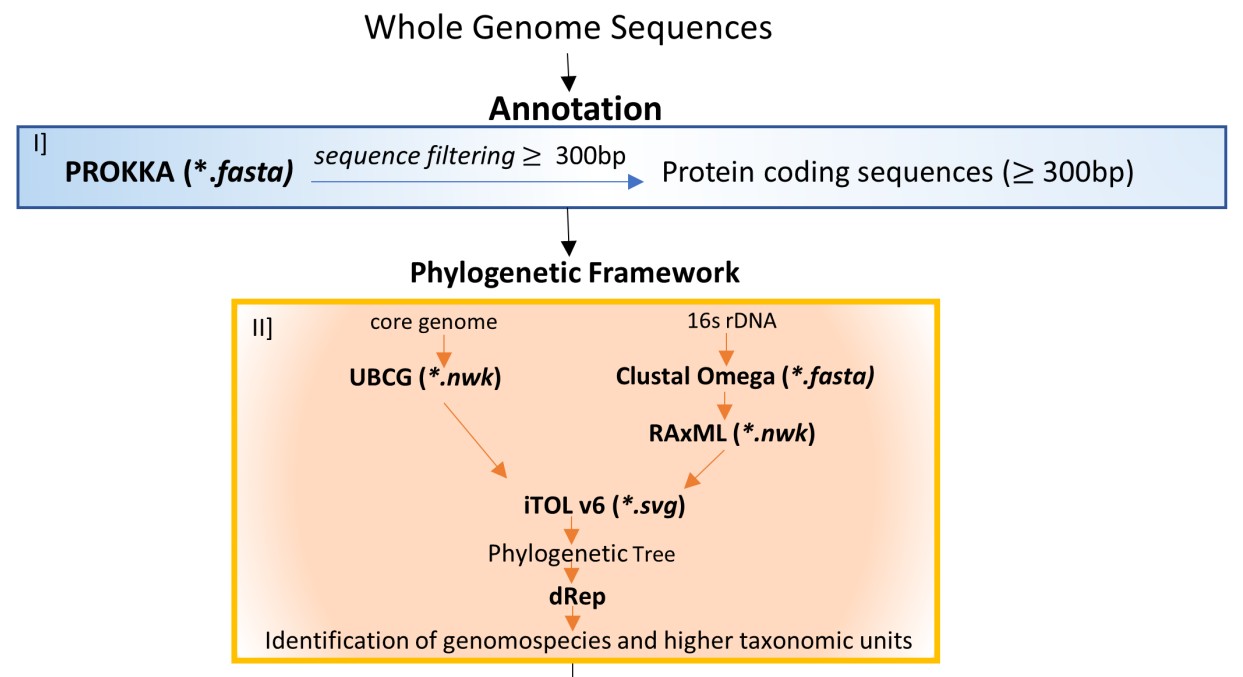

**FIG 2** Overview of the pipeline used for the detection and verification of HGT genes: (I) protein-coding DNA sequences ≥ 300 bp were extracted; (II) definition of genomospecies, genera, and families using comparative analysis of draft genome sequences and genes coding for 16S rDNA. The definitions were reconciled using inferred phylogenesis in Fig. 1; Fig. S1; (III) identification of nearly identical genes co-shared by at least two different genomospecies, genera, families, and phyla followed by statistical verification. Genuine genes associated with HGT were visualized by network analysis.

enrich the gut microbiome and to culture novel bacterial isolates, animals were reared in commercial as well as backyard farms, were of different breeds, ages, and sex, and were fed with food supplemented with probiotics (summarized in Table S1). The less stringent conditions thus allowed us to detect both recent and past HGT events, as HGT genes have a tendency to be adapted toward the genome of novel bacteria host (31, 32) to be highly expressed in the cell (33).

## Genes shared by different genomospecies

Horizontally acquired genes are predominantly exchanged between closely related taxonomical units, whereas the frequency of HGT is decreasing with the phylogenetic distance due to limitations for MGE (34). We therefore first defined species boundaries

based on available genomic sequences by dRep. Genome sequences with ≥95% nucleotide identity were considered the same genomospecies (35). Whole-genome sequences were binned into seven phylogenetically distant groups such as *Actinobacteria* (included one group), *Bacteroidetes* (included one group), *Proteobacteria* (included one group), genomes of *Firmicutes* members were split into four groups, one group was co-shared with genomes of *Fusobacteria*. dRep comparison was then applied separately for every group. Although dRep was initially designed for dereplication and comparison of metagenomic assembled genomes (35), the Mash-distance clustering (part of dRep) of individually sequenced *Firmicutes* genomes from our data set proved unsuccessful. As a result, we opted to enhance accuracy by categorizing genomes into distinct groups based on the branching pattern of the UBCG tree (Fig. 3, depicted as an unrooted tree). Finally, in defining genomospecies using the UBCG tree, 16S rDNA, and dRep clustering, they exhibited concordance. In addition, the GTDB taxonomy mostly aligned with these results (Table S1). To identify nearly identical genes shared by different genomospecies, we first determined a non-redundant gene pool (non-redundant pan-genome, NRPG) for every genomospecies.

On the contrary, redundant genes were considered genes with ≥99% nucleotide identity over ≥99% gene length shared by the same operational taxonomic unit, herein the same genomospecies. Only one such gene was retained while the others were discarded from the (species) pan-genome.

Among the 1,235,343 protein-coding sequences ≥300 bp present in 452 genomes, we identified a total of 694,388 genes (the sum of all genomospecies NRPGs) across all 249 distinct genomospecies. Within these genes, 10,629 were unique and non-redundant, shared by at least two genomospecies, specifically identified in 231 out of the 249 genomospecies examined (Table 1) averaging 110 genes shared per genomospecies. In general, as expected the number of shared genes with members of the same family and phylum was bigger than a number of shared genes shared with members of other families and phyla, respectively (Fig. 4) (34). Eighteen genomospecies did not co-share any gene at all (Fig. 4):

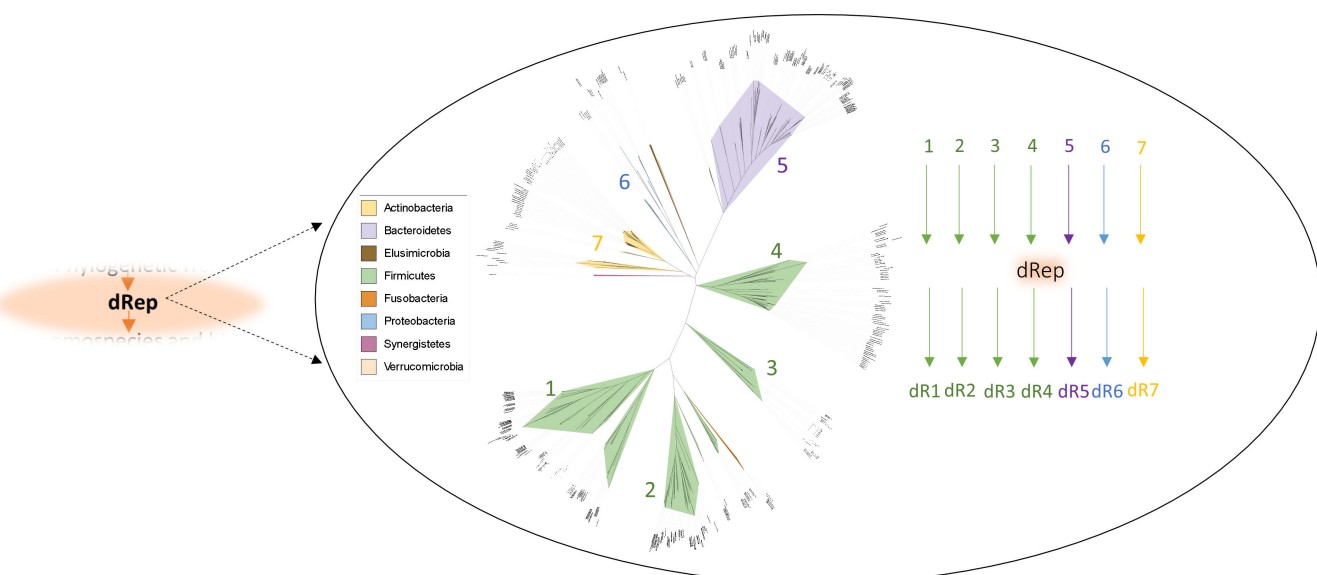

**FIG 3** An unrooted phylogenetic tree generated through UBCG analysis was employed to categorize the genome sequences into seven distinct phylogenetically distant groups: *Lactobacillales* (*Firmicutes*, group1), *Lachnospirales* (*Firmicutes*, group2), *Negativicutes* (*Firmicutes*, group3), *Oscillospirales* (*Firmicutes*, group4), *Bacteroidetes* (group 5), *Proteobacteria* (group 6), and *Actionobactera* (group7). Within each of these groups, the genome sequences were subjected to comparison using dRep. The objective of this method was to define clusters of genomes categorized as the same genomospecies, identified as a group of isolates with an average nucleotide identity (ANI) of ≥95%. The results of dRep clustering were also reconciled during the identification of more distant related taxa, such as genera and families. Without subgrouping, the analysis of *Firmicutes* failed due to the polyphyletic nature of the phylum. In addition, sub-grouping also enhanced and expedited the comparison.

1.  *Cloacibacillus* sp. (*GTDB An23 sp002159945, Synergistaceae*)

2.  *Cutibacterium acnes* (*C. acnes, Propionibacteriaceae*)

3.  *Cellulomonas cellasea* (*Cellulomonas* sp., *Cellulomonadaceae*)

4.  *Actinomyces viscosus* (*Actinomyces oris, Actinomycetaceae*)

5.  *Schaalia hyovaginalis* (*Pauljensenia hyovaginalis, Actinomycetaceae*)

6.  *Bifidobacterium ruminantium* (*B. ruminantium, Bifidobacteriaceae*)

7.  *Sutterella massiliensis* (*Sutterella* sp., *Burkholderiaceae*)

8.  *Elusimicrobium* sp. (*UBA1436 sp002159705, Elusimicrobiaceae*)

9.  *Akkermansia muciniphila* (*A. muciniphila, Akkermansiaceae*)

10. *Oscillibacter valericigenes* (*Oscillibacter ruminantium, Oscillospiraceae*)

11. *Veillonella magna* (*Veillonella_A magna, Veillonellaceae*)

12. *Paraclostridium benzoelyticum* (*P. benzoelyticum, Peptostreptococcaceae*)

13. *Clostridium butyricum* (*C. butyricum, Clostridiaceae*)

14. *Oceanobacillus oncorhynchi* (*O. oncorhynchi, Amphibacillaceae*)

15. *Bacillus aerophilus* (*Bacillus altitudinis, Bacillaceae*)

16. *Bacillus licheniformis* (*B. licheniformis, Bacillaceae*)

17. *Enterococcus hirae* (not assigned, *Enterococcaceae*)

18. *Enterococcus gallinarum* (*Enterococcus_D gallinarum, Enterococcaceae*)

The species of *Bacteroidaceae* belonged to the main contributors of shared genes, especially, *Bacteroides gallinaceum* (*Phocaeicola sp002161565*) with 1,508 shared genes, *Bacteroides caecigallinarum* (*Phocaeicola* sp.) with 1,000 shared genes, *Phocaeicola barnesiae* (*P. barnesiae*) with 755 shared genes, *Bacteroides caecigallinarum* (*Phocaeicola sp900066445*) with 746 shared genes, and *Bacteroides ovatus* (*B. ovatus*) with 724 shared genes. Among *Firmicutes*, the biggest contributor was genomospecies *Flavonifractor* sp. (*Flavonifractor sp002161085, Oscillospiraceae*) with 590 shared genes, among *Actinobacteria Gordonibacter* sp. (*Rubneribacter sp002159915, Eggerthellaceae*) with 260 shared genes, among *Proteobacteria Desulfovibrio* sp. (*Desulfovibrio sp002159665, Desulfovibrionaceae*), and *Desulfovibrio piger* (*Desulfovibrio sp900556755*) with 13 shared genes and finally among *Fusobacteria Fusobacterium mortiferum* (*Fusobacterium_A mortiferum*) with 63 shared genes.

To investigate how individual isolates contribute to the number of shared genes and if and how the co-shared gene pool is influenced by the original source of bacteria, we backtracked the genomospecies-shared genes to genomes of all isolates (Fig. S2 and S3; Fig. 4). We detected extensive gene transfer between chicken and porcine bacteria (Fig. S2 and Table S1) (see NCBI projects PRJNA377666 and PRJNA658263 for more details). We therefore assume that MGEs are widely exchanged in a time-short manner across different environments. Regarding the number of shared genes, most isolates of the same genomospecies contribute to the shared gene pool to the same extent and genomes of the isolates may or may not bear the same genes (Fig. 4; Fig. S2). On the other hand, substantial differences (> order of magnitude) in the number of shared genes were found among isolates of genomospecies: *Alistipes sp900290115* (isolates SAMN34359385, SAMN06473718),

**TABLE 1** Summary of the number of genes present in all genomes, in non-redundant pangenomes of all genomospecies (NRPG gs), and numbers of genes co-shared by different genomospecies, genera, families, and genes suspected of HGT were primarily determined from the NRPG gs group

|  | All genomes | NRPG gs | Genes suspicious to HGT | | | | |
|---|---|---|---|---|---|---|---|
|  | Isolates | Genomospecies | Genomospecies | Genera_16S | Families_16S | Families | Phyla |
| Total number of examined taxa | 452 | 249 | 249 | 138 | 85 | 33 | 8 |
| Identified genes | 1,234,694 | 694,388 | 10,629 | 6,545 | 5,655 | 2,314 | 888 |
| Genes of unknown function | 161,066 | 51,523 | 1,003 | 618 | 521 | 140 | 38 |
| Genes with predicted function | 1,073,628 | 642,865 | 9,626 | 5,927 | 5,134 | 2,174 | 850 |
|  | (86.95%) | (92.58%) | (90.56%) | (90.56%) | (90.79%) | (93.95%) | (95.72%) |

*Butyricicoccus pullicaecorum* (SAMN06473643, SAMN06473645), *Streptococcus alactolyticus* (SAMN15872590, SAMN14913642, SAMN15872592), and *Ligilactobacillus salivarius* (SAMN14913633, SAMN15872553, SAMN34359433, SAMN14913565, SAMN34359395, SAMN06473764, SAMN34359446, SAMN34359409, SAMN34359391, SAMN14913613, SAMN06473742, SAMN06473601).

## Identification and statistical verification of HGT gene pools

To determine whether co-shared genes are really associated with MGEs and HGT, genes were functionally classified into clusters of orthologous groups (COG ID), that is, a group of genes with the same protein function, belonging to COG categories. Moreover, gene pools co-shared by different genera, families, and phyla were also defined, respectively.

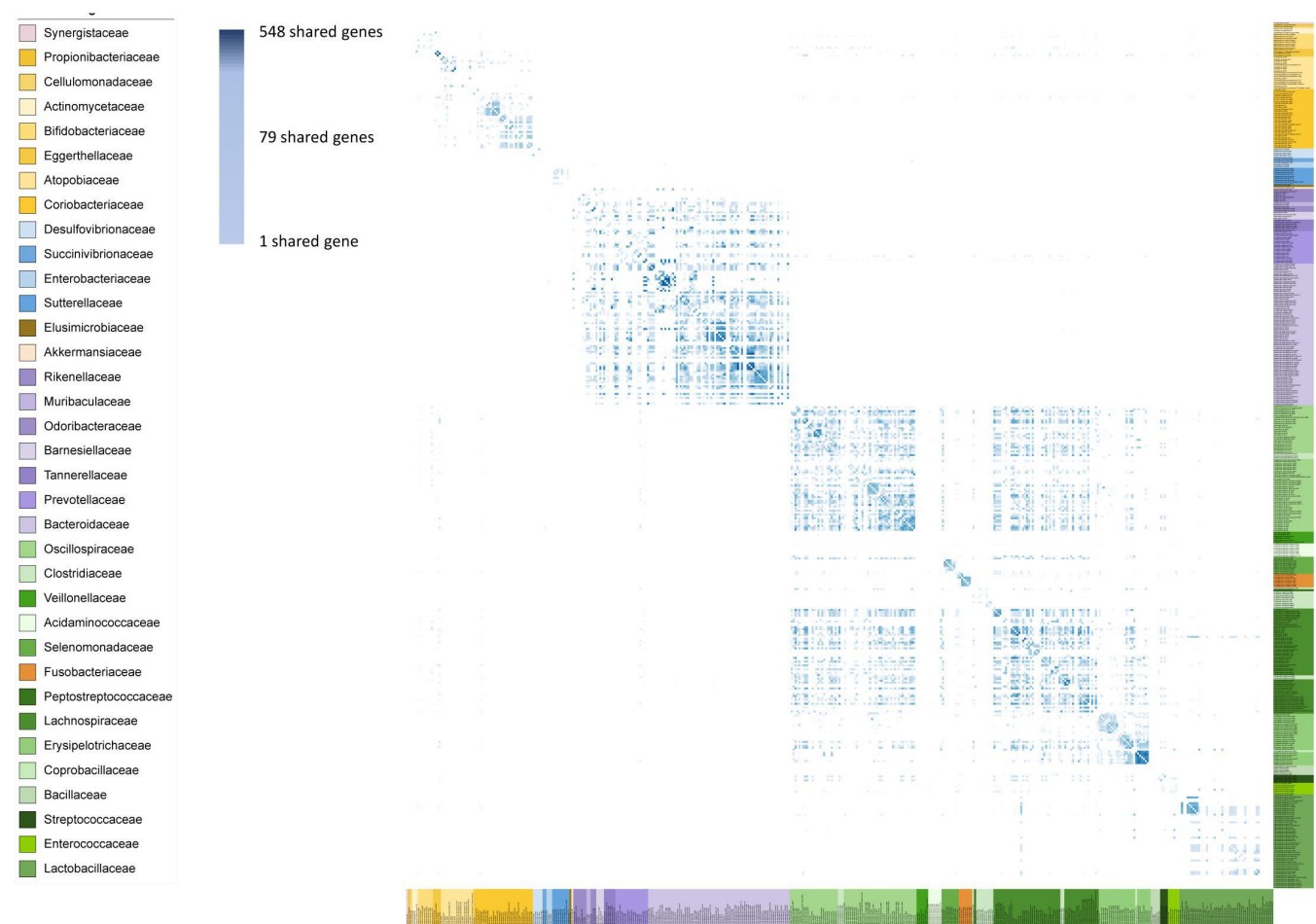

**FIG 4** Heatmap showing the abundance of genes co-shared by two different genomospecies backtracked to individual genomes. Mind that we used a logarithmic-scale color bar. Individual genomes (coloring as in Fig. 1) were ordered according to the UBCG tree. For more details, see Fig. S3.

Whereas genus definition based on the comparison of whole-genome sequences has not yet been properly determined, we applied the classification using comparative analysis of genes encoding 16S rDNA. As we have shown earlier, the UBCG and 16S rDNA trees were congruent in branching. Isolates with ≥98.5% identity in the 16S rDNA belonged to the same genomospecies, as proposed earlier (36). For genus (Genera_16S) and family (Families_16S) definitions, 94.5% and 92% identity thresholds were applied, respectively (37). We also retain isolates belonging to the same family (Families) and phylum (Phyla) based on the closest hit to the NCBI RefSeq16S rRNA sequence database (see Materials and Methods section). Again, for every operational taxonomic unit, a pan-genome was determined, and then genes co-shared by different operational taxonomic units were identified and functionally characterized (Table 1). Among 138 genera, 6,545 genes were co-shared by at least two different genera, 5,656 genes were co-shared across 85 families (Families_16S), 2,315 genes were co-shared across 33 taxonomic families (Families), and 888 genes were co-shared across eight phyla. Based on the definition of Families_16S, several corrections in nomenclature had to be made to be consistent with 16S rDNA comparison (Table S1): *Clostridioides difficile* was classified as *Clostridiaceae*, *Coprobacillus cateniformis* was clustered among *Erysipelotrichaceae*, *Mordavella massiliensis* isolates (SAMN14913548, SAMN14913570, SAMN14913587) belonged to *Erysipelotrichaceae*, and *Eubacterium* sp. (SAMN14913587, SAMN14913587) belong to *Lachnospiraceae*. Hence, the number of families was 33 in the analysis. We are aware that boundaries of operational taxonomic units cannot be precisely defined for all members of the same taxonomic units because evolutionary constraints vary.

Next, we assumed that vertically passed genes, such as those involved in amino (COG category E), and nucleotide (F) metabolism and transport, translation (J), transcription (K), DNA replication (L) will not be preserved in the set of genuine HGT genes, and on the contrary genes involved in DNA recombination processes (L), intracellular trafficking and secretion (U), and defense mechanisms (V) will be enriched. The distributions of COG categories will thus differ in an average bacterial genome and the set of genuine HGT genes. This hypothesis finds support in the findings reported by Kloub et al. (38). Therefore, we compared the distributions of COG categories in different gene pools (Fig. 5). We assume that in terms of relative frequency of different COG categories, the gene pool of all genomes represents an average genome with a small number of genes associated with HGT. On the contrary, the gene pool co-shared by different phyla predominantly includes genes prone to HGT, therefore also COG categories associated with HGT. However, this gene pool is incomplete due to mechanical limitations in the transfer of foreign DNA between phylogenetically distant bacteria (34). Non-parametric Friedman test confirmed that statistical differences exist across different gene pools ($P$-values < 0.01). *Post hoc* Dunn's test with Bonferroni correction revealed no significant difference between the group of all genes and genomospecies co-shared genes. Indeed, this finding is in line with the presence of genes coding for ribosomal proteins and other phylogenetically conserved proteins in the genomospecies gene pool which are supposed to be single-copy genes used for phylogenetic reconstruction (26). Statistical significance was detected between group of all genes and genera co-shared genes ($P$ < 0.05) and between the group of all genes and Families_16S, Families, and Phyla co-shared genes (all $P$ < 0.01), so the composition of the genera-co-shared gene pool has been shifted from the average genome, especially genes involved in basic metabolic and cell processes have been missing in the gene pool. Therefore, nearly identical genes co-shared by different genera were considered mobilized genes *via* HGT. Similar settings were also identified or applied elsewhere (29, 39) and more importantly, probabilistic approaches to detect HGT also supported the assumption that identical genes shared by different genera were likely horizontally transferred between bacteria (40).

Since genera names are not provided systematically and up-to-date either by GTDB or NCBI (e.g., strain An23, submitted by us to NCBI in 2017, now bears the genus name "*An23*" in GTDB, as of 21 August 2023), we assessed the identifier based on the name of NCBI family, optionally with a number suffix (*Lachnospiraceae*, *Lachnospiraceae_1* to

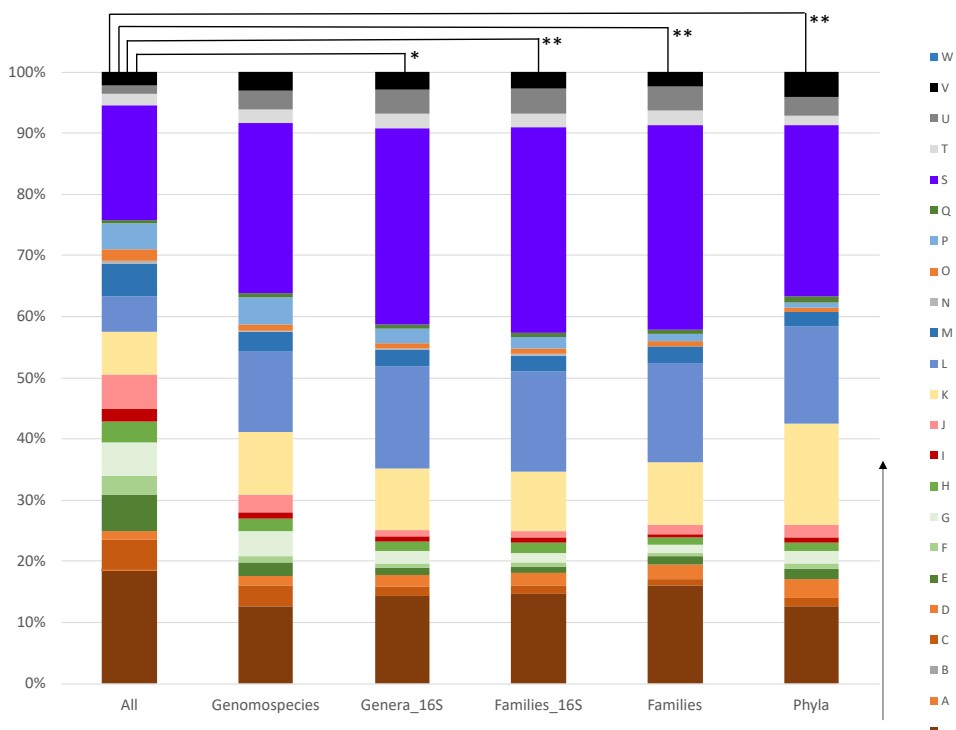

**FIG 5** Cluster of orthologous groups (COG) distribution of genes (≥300 bp) present in all investigated genomes and in a subset of genes shared among diverse genomospecies, genera (as determined through 16S rDNA analysis and conventional methods), families, and phyla. Dunn's *post hoc* test was utilized for the statistical analysis of COG profiles present in the shared gene pools of a determined taxonomic level vs a common bacterial genome profile (category "All"). (-) COG not assigned; (A) RNA processing and modification; (B) chromatin structure and dynamics; (C) energy production and conversion; (D) cell cycle control, cell division, and chromosome partitioning; (E) amino acid metabolism and transport; (F) nucleotide metabolism and transport; (G) carbohydrate metabolism and transport; (H) coenzyme metabolism; (I) lipid metabolism; (J) translation; (K) transcription; (L) replication and repair; (M) cell wall/membrane/envelop biogenesis; (N) cell motility; (O) post-translational modification, protein turnover, chaperone functions; (P) inorganic ion transport and metabolism; (Q) secondary structure; (S) function unknown; (T) signal transduction; (U) intracellular trafficking and secretion; (V) defense mechanism; (W) extracellular structures; Dunn's *post hoc* test: *$P < 0.05$; **$P < 0.01$.

*Lachnospiraceae_20*, i.e., 21 genera). Isolates of the same genera keep the same name (Table S1).

In total, 6,545 unique genes were co-shared across different genera (Table S1; Fig. 6A) with an average of 126 genes shared per genus. Whereas 13 genera did not co-share any gene, main contributors to the shared gene pool comprise *Bacteroidaceae_17* (according to GTDB *Phocaeicola, Bacteroidaceae*; 1,508 shared genes), *Oscillospiraceae_18* (*UBA9475*, early *Pseudoflavonifractor, Oscillospiraceae*, 999 genes), *Bacteroidaceae_19* (*Phocaeicola, Bacteroidaceae*, 975 genes), *Bacteroidaceae_22* (*Phocaeicola, Bacteroidaceae*, 755 genes), and *Lachnospiraceae_20* (*Mediterraneibacter, Lachnospiraceae*, 632 genes). Among *Fusobacteria*, the biggest shared-gene contributor was *Fusobacteriaceae_0* (*Fusobacterium_B, Fusobacteriaceae*, 11 genes), among *Proteobacteria Desulfovibrionaceae_0* (*Desulfovibrio, Desulfovibrionaceae*, 13 genes) and among *Actinomyceta Eggerthellaceae_1* (*Gordonibacter, Eggerthellaceae*, 112 genes). We can argue that we identified an unexpected number of shared genes in some genera and different genera in this manuscript comprise the same genus, for example, *Phocaeicola* according to GTDB taxonomy. So, we should be very careful to consider such shared genes as mobilized genes. However, members of *Bacteroidaceae* shared 1,272 nearly identical genes with other members of other families, followed by members of *Lachnospiraceae* with 806 shared genes, *Oscillospiraceae* with 728 shared genes, and *Prevotellaceae* with 636 shared genes.

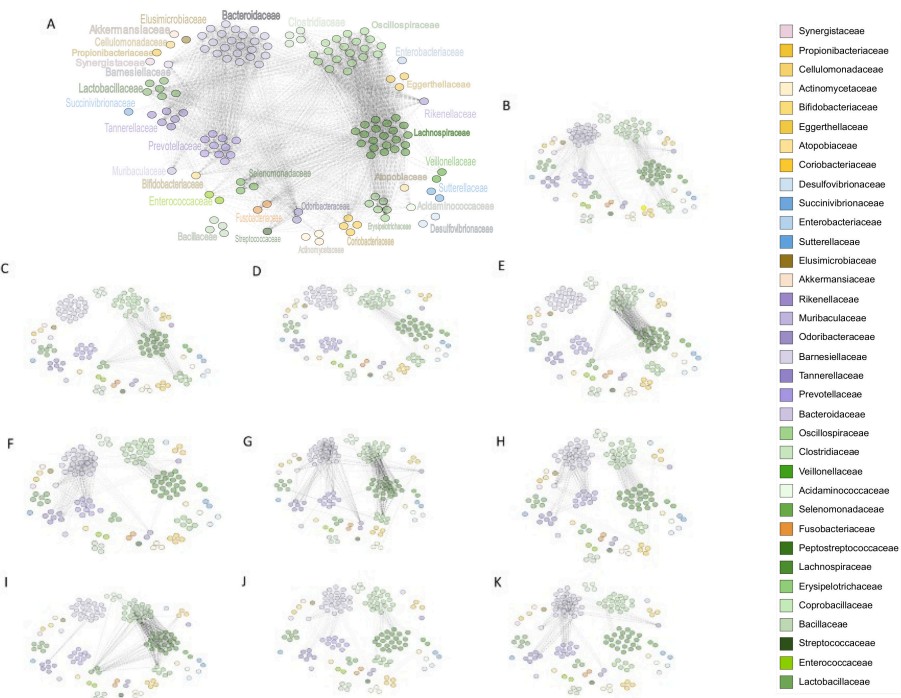

**FIG 6** Network visualization of identified HGT genes, genes co-shared by at least two different genera, includes. Nodes represent different genera, whereas edges are HGT genes. Genera names are depicted in panel A, all other panels use the same network of nodes. (A) All genuine HGT genes (*n* = 6,545); (B) genes with no functional annotation (*n* = 618); (C) 2C5R4—Transposon-encoded protein TnpV (*n* = 22); (D) 28HUM—TrbL/VirB6 plasmid conjugal transfer protein (*n* = 17); (E) 28NID—cysteine-rich VLP domain-containing protein; transcriptional regulator (*n* = 24); (F) COG0358—DNA primase DnaG (*n* = 68); (G) COG0582 —Integrase/recombinase, phage integrase FimB (*n* = 133); (H) COG1192—ParA ATPase involved in plasmid-prophage partitioning (*n* = 52); (I) COG1961— site-specific DNA recombinase SpoIVCA/DNA invertase PinE (*n* = 70); (J) COG3505—Type IV secretory pathway, VirD4 component, TraG/TraD family ATPase (*n* = 52); (K) COG4974—site-specific recombinase XerD (*n* = 93); The transparency of the edges indicates the number of transferred genes.

## Characterization and network analysis of HGT genes

To characterize HGT genes and to assess their function in a cell, genes were functionally annotated (Table S2). For 90.56% of genes, COG ID and COG category were assigned, whereas 9.4%, 618 are genes of unknown function (Fig. 6B). Compared to the gene set of an averaged genome, HGT genes shared by different genera were reduced (>2-fold) in COG categories: energy production and conversion (C), amino acid metabolism and transport (E), nucleotide metabolism and transport (F), carbohydrate metabolism and transport (G), coenzyme metabolism (H), lipid metabolism (I), translation (J), cell motility (N), post-translational modification, protein turnover, chaperone functions (O), in terms of relative frequency (Table S2). Such genes and functions are usually not associated with HGT. On the contrary, genes present in COG categories: intracellular trafficking and secretion (U) and replication and repair (L) were enriched (>2-fold), especially genes coding for proteins involved in DNA repair such as recombinases/transposases/integrases, post-translational modification, and type IV secretory pathway, followed by virulence factors, anti-restriction proteins, phage and retron-related genes, and transcriptional response regulators. Of note, HGT genes shared by different families and phyla are also enriched in COG categories L and U, and HGT genes shared by phyla are also in categories: cell cycle control, cell division, chromosome partitioning (D), transcription (K), and secondary structure (Q). The list of most common COGs across HGT genes co-shared across genera is summarized in Table 2.

Next, we focused on the network analysis of selected common COGs: COG0582 (integrase/recombinase; COG category L; Fig. 6G), COG4974 (site-specific recombinase

**TABLE 2** Characterization of the most prevalent COGs within the identified HGT genes

| Number of genes within the COG | COG ID | COG category | Gene/protein description |
| --- | --- | --- | --- |
| 133 | COG0582 | LX | Integrase/recombinase includes phage integrase FimB |
| 93 | COG4974 | L | Site-specific recombinase XerD |
| 70 | COG1961 | L | Site-specific DNA recombinase SpoIVCA/DNA invertase PinE |
| 68 | COG0358 | L | DNA primase DnaG |
| 52 | COG3505 | U | Type IV secretory pathway, VirD4 component, TraG/TraD family ATPase |
| 52 | COG1192 | DN | ParA-like ATPase involved in P1 plasmid-prophage partitioning |
| 45 | COG1309 | K | Multidrug efflux transporter transcriptional repressor AcrR |
| 45 | COG3843 | U | Type IV secretory pathway, VirD2 component (relaxase) |
| 41 | COG1131 | V | ABC-type multidrug transport system, ATPase component |
| 40 | COG3451 | U | Type IV secretory pathway, conjugation system ATPase |
| 37 | COG0745 | KT | DNA-binding transcriptional dual regulator OmpR |
| 34 | COG4227 | L | Anti-restriction protein ArdC |
| 33 | COG1196 | D | Chromosome segregation ATPase Smc |
| 32 | COG3344 | X | Retron-type reverse transcriptase YkfC |
| 32 | COG3943 | S | Uncharacterized protein RhuM, Salmonella virulence factor |
| 31 | COG0642 | T | Signal transduction histidine kinase |
| 30 | COG1484 | L | DNA replication protein DnaC |
| 29 | COG1132 | V | ABC-type multidrug transport system, ATPase, and permease component |
| 28 | COG0550 | L | DNA topoisomerase IA TopA |
| 24 | 28HNW | | Conjugative transposon TraM protein |
| 24 | 28NID | | Cysteine-rich VLP domain-containing protein; transcriptional regulator |
| 23 | COG3701 | U | Type IV secretory pathway, conjugative transposon TraK protein |
| 23 | COG0467 | T | RecA-superfamily ATPase, KaiC/GvpD/RAD55 family |
| 23 | COG0500 | QR | SAM-dependent methyltransferase SmtA |
| 22 | 28IE2 | | Conjugative transposon TraJ protein |
| 22 | 2CI0Q | | Protein of unknown function (DUF4099) |
| 22 | 2C5R4 | | Transposon-encoded protein TnpV |
| 21 | COG3547 | X | Transposase |
| 21 | COG3772 | M | Phage-related lysozyme (muramidase), GH24 family |
| 21 | COG4584 | X | Transposase |
| 20 | COG3504 | U | Type IV secretory pathway, conjugative transposon TraN protein |
| 20 | COG3935 | L | DNA replication protein DnaD, phage replisome organizer |
| 20 | COG2204 | T | DNA-binding transcriptional response regulator, NtrC family |
| 20 | COG3385 | X | IS4 transposase InsG |
| 19 | 2D42G | | Helix-turn-helix domain; excisionase Xis |
| 19 | COG0454 | KR | N-acetyltransferase, GNAT superfamily (includes histone acetyltransferase HPA2) PhnO |
| 19 | COG1373 | R | Predicted ATPase, AAA + superfamily |
| 18 | COG4474 | X | Uncharacterized SPBc2 prophage-derived protein YoqJ |
| 18 | COG4734 | V | Anti-restriction protein ArdA |
| 18 | COG5314 | X | Conjugal transfer/entry exclusion protein |
| 17 | COG1136 | M | Bacteriocin export ABC transporter, lactococcin 972 group |
| 17 | 28JHB | | Conjugative transposon protein TraO |
| 17 | COG0863 | L | DNA adenine methyltransferase |
| 17 | 293NS | | Domain of unknown function (DUF4133) |
| 17 | 28HUM | | TrbL/VirB6 plasmid conjugal transfer-like protein |
| 15 | COG0655 | C | Multimeric flavodoxin WrbA, includes NAD(P)H:quinone oxidoreductase |
| 15 | COG3039 | X | Transposase and inactivated derivatives, IS5 family |
| 15 | COG4804 | R | Restriction endonuclease-like (RecB) superfamily, DUF1016 family |
| 14 | COG0827 | L | Adenine-specific DNA N6-methylase YtxK, DNA restriction-modification system |
| 14 | 28M8P | | Domain of unknown function, DUF3872 family |
| 14 | 28KU3 | | PcfK-like family protein |

(*Continued on next page*)

**TABLE 2** Characterization of the most prevalent COGs within the identified HGT genes (*Continued*)

| Number of genes within the COG | COG ID | COG category | Gene/protein description |
|---|---|---|---|
| 14 | COG2452 | X | Predicted site-specific integrase-resolvase 3ILX |
| 13 | 2F5RM | | Domain of unknown function (DUF4134) |
| 13 | COG2253 | V | Nucleotidyl transferase AbiEii toxin, Type IV TA system, viral defense |
| 13 | 28JQ1 | | PcfJ domain-containing protein |
| 13 | 296J0 | | Plasmid mobilization relaxosome protein MobC |
| 13 | 28I1B | | Protein of unknown function (DUF3801) |
| 13 | COG1193 | L | dsDNA-specific endonuclease/ATPase MutS2 |
| 13 | COG1479 | V | DNAse/DNA nickase specific for phosphorothioated or glycosylated phage DNA |
| 13 | COG2826 | X | Transposase and inactivated derivatives, IS30 family |
| 13 | COG3436 | X | Transposase |
| 12 | 2E51N | | Bacterial mobilization protein MobC, the group of relaxases |
| 12 | COG0389 | L | Nucleotidyltransferase/DNA polymerase DinP involved in DNA repair |
| 12 | COG3315 | Q | O-Methyltransferase involved in polyketide biosynthesis YktD |
| 12 | COG1191 | K | DNA-directed RNA polymerase specialized sigma28 subunit FliA, involved in motility |
| 11 | COG1106 | R | AAA domain, putative AbiEii toxin, Type IV TA system |
| 11 | COG1409 | T | 3′,5′-cyclic AMP phosphodiesterase CpdA |
| 11 | 2BWP0 | | Domain of unknown function (DUF1896) |
| 11 | COG1073 | T | Fermentation-respiration switch esterase FrsA, DUF1100 family |
| 11 | COG1533 | L | DNA repair photolyase SplB, spore photoproduct lyase |
| 11 | COG0577 | V | ABC-type antimicrobial peptide transport system, permease component |
| 11 | COG3177 | K | Fic family protein |
| 11 | COG4200 | V | Predicted lantabiotic-exporting membrane pepmease, EfiE/EfiG/ABC2 family |
| 10 | 28HUZ | | Domain of unknown function (DUF4366) |
| 10 | COG1277 | O | Motility-associated transport system permease protein |
| 10 | 2DR7C | | Protein of unknown function (DUF3408) |
| 10 | 2E6×0 | | Protein of unknown function (DUF3408) |
| 10 | COG1349 | KG | Replication initiator protein A (RepA) N-terminus |
| 10 | 28KSX | | RteC protein; tetracycline resistance (Tcr) elements |
| 10 | 2C2YH | | T-DNA endonuclease VirD1; plasmid mobilization relaxosome protein MobC |
| 10 | COG0286 | V | Type I restriction-modification system, DNA methylase subunit HsdM |
| 10 | COG0457 | R | Tetratricopeptide (TPR) repeat |

XerD; L; Fig. 6K), COG1961 (site-specific recombinase SpoIVCA/DNA invertase PinE; L; Fig. 6I), COG0358 (DNA primase DnaG; L; Fig. 6F), COG3505 (type IV secretory pathway, VirD4 component; U; Fig. 6J), COG1192 (parA-like ATPase involved in chromosome/plasmid partitioning; D; Fig. 6H) with 133, 93, 70, 68, 52, and 52 different genes, respectively. In the analysis, we also included common genes with assigned COG category S, unknown function: 2C5R4 (transposon-encoded protein TnpV; S; Fig. 6C), 28HUM (TrbL/VirB6 plasmid conjugal transfer like protein; S; Fig. 6D), 28NID (cysteine-rich VLP domain-containing protein; transcriptional regulator; S; Fig. 6E) with 24, 22, and 17 different gene variants, respectively. Except for DnaG, all other proteins are associated with HGT including cysteine-rich proteins (41).

Network analysis (Table 3) showed that the greatest diversity (i.e., network radius) was found among 30 genes of unknown function. In general, genes are predominantly co-shared by the members of the same family, especially within *Firmicutes* and *Bacteroidetes*. Whereas different genes encoding recombinase XerD have been mainly transferred between members of *Bacteroidetes*, genes encoding recombinase SpoIVCA have been disseminated among *Firmicutes*. Common COGs associated with conjugal (2C5R4, 28HUM) or phage (28NID) transfer have been found only in *Firmicutes*. If some COGs (COG0358, COG0582, COG1192, and COG3505) were shared among *Bacteroidetes* and *Firmicutes*, different gene variants (network edges) were found in the particular phylum. Based on the result of network visualization, we assume that

**TABLE 3** Network analysis of all and selected HGT genes depicted in Fig. 6 and inferred by Cytoscape v3.9.0

| | All HGT genes (*n* = 6545) | HGT genes of unknown function (*n* = 618) | 2 C5R4 (*n* = 22) | 28HUM (*n* = 17) | 28NID (*n* = 24) | COG 0358 (*n* = 68) | COG 0582 (*n* = 133) | COG 1192 (*n* = 52) | COG 1961 (*n* = 70) | COG 3505 (*n* = 52) | COG 4974 (*n* = 93) |
|---|---|---|---|---|---|---|---|---|---|---|---|
| Number of nodes | 138 | 138 | 138 | 138 | 138 | 138 | 138 | 138 | 138 | 138 | 138 |
| Number of edges | 3323 | 1246 | 204 | 104 | 384 | 380 | 620 | 340 | 622 | 174 | 254 |
| Avg. number of neighbors | 26.397 | 9.267 | 7.769 | 4.727 | 12 | 7.611 | 9 | 7.077 | 14.85 | 3.556 | 7.355 |
| Network diameter | 5 | 7 | 5 | 5 | 5 | 4 | 5 | 5 | 4 | 5 | 5 |
| Network radius | 3 | 4 | 3 | 3 | 2 | 2 | 3 | 3 | 2 | 3 | 3 |
| Characteristic path length | 2.227 | 2.723 | 2.074 | 2.147 | 1.728 | 2.054 | 2.144 | 2.034 | 1.695 | 2.556 | 2.105 |
| Clustering coefficient | 0.746 | 0.533 | 0.721 | 0.604 | 0.737 | 0.653 | 0.626 | 0.707 | 0.703 | 0.42 | 0.527 |
| Network density | 0.22 | 0.157 | 0.311 | 0.225 | 0.387 | 0.217 | 0.257 | 0.283 | 0.381 | 0.137 | 0.245 |
| Network heterogeneity | 0.535 | 0.796 | 0.634 | 0.66 | 0.558 | 0.747 | 0.819 | 0.673 | 0.496 | 0.782 | 0.684 |
| Network centralization | 0.327 | 0.381 | 0.4 | 0.486 | 0.482 | 0.466 | 0.363 | 0.517 | 0.49 | 0.351 | 0.522 |
| Connected components | 16 | 33 | 112 | 117 | 107 | 82 | 64 | 91 | 88 | 88 | 97 |

strong selection constraints on gene sequences exist at the phylum level to adapt to different mechanisms of HGT.

Finally, we also detected 15 genes present in both Gram-positive and Gram-negative bacteria (Table 4). Besides functional analysis based on eggNOG-mapper, corresponding protein sequences were further characterized using blastp (42) against the comprehensive antibiotic resistance database (43) and/or non-redundant protein sequences in NCBI (44). These genes included genes associated with antibiotic resistance, toxicity, mobilization, and defense mechanisms against pathogens, genes that have also been suspected of HGT earlier (43, 45, 46), as well as a gene of unknown function.

Surprisingly, only a small fraction (30 out of 6,545) of genes thought to be engaged in HGT have been linked to acquired resistance mechanisms (Table S2). To establish a comparison with the findings presented in references (31, 47), we directed our attention to the tetracycline resistance genes. In our HGT collection, we have identified 14 distinct gene/gene variants encompassing alleles of *tet (40)*, *tet (44)*, *tet(M)*, *tet(O)*,

**TABLE 4** HGT genes co-shared by Gram-positive and Gram-negative bacteria

| gene_ID | COG ID | COG category | Definition |
|---|---|---|---|
| gene_396 | COG0500 | Q | RlmA(II) methyltransferase |
| gene_623 | 2DB8K | G | Streptomycin adenylyltransferase |
| gene_645 | COG0480 | J | Tetracycline resistance protein Tet(O) |
| gene_814 | COG0480 | J | Tetracycline resistance protein Tet(W) |
| gene_941 | COG2820 | F | Uridine phosphorylase |
| gene_942 | COG1708 | S | Aminoglycoside nucleotidyltransferase ANT (9) |
| gene_943 | COG3677 | L | IS1595-like element ISSag10 family transposase |
| gene_944 | COG0617 | J | Lincosamide nucleotidyltransferase LnuC |
| gene_4123 | COG0500 | Q | SAM-dependent methyltransferase |
| gene_4124 | COG2206 | T | CRISPR-associated endonuclease Cas3-HD |
| gene_9378 | - | - | Plasmid replication protein |
| gene_9670 | COG0645 | S | CRISPR/Cas system-associated protein Cas3 |
| gene_2789 | COG0500 | H | RlmA(II) methyltransferase |
| gene_10615 | - | - | Hypothetical protein |
| gene_10616 | - | - | Trypsin-like serine protease |

**TABLE 5** Summary on the evaluation of HGT prediction in the genome of *Phocaeicola sp900066445* 1_COKtk using different computational tools

|  | DHGT-ComAnalysis-100id[a] | DHGT-ComAnalysis-99id[b] | MetaCHIP | ShadowCaster | AlienHunter | HGTector |
|---|---|---|---|---|---|---|
| DHGT-ComAnalysis-100id | - | *43[c]* | *17* | 25 | 34 | 18 |
| DHGT-ComAnalysis-99id | *30* | - | *25* | 32 | 43 | 18 |
| MetaCHIP | *17* | *25* | - | 38 | 52 | 9 |
| ShadowCaster | 25 | 32 | 38 | <u>-</u> | <u>191[d]</u> | <u>156</u> |
| AlienHunter | 34 | 43 | 52 | <u>191</u> | <u>-</u> | <u>135</u> |
| HGTector | 18 | 18 | 9 | <u>156</u> | <u>135</u> | <u>-</u> |
| **Total number of predicted HGTs** | **74** | **77** | **89** | **511** | **633** | **437** |

[a]DHGT_ComAnalysis; acronym for the pipeline purposed in this study derived from the title: **D**etecting **H**orizontal **G**ene **T**ransfer through **Com**parative **Analysis.**
[b]99id of 100id indicate if 99% or 100% identity settings were applied in the analysis.
[c]Composition-based methods are in italics.
[d]Methods using "best match" and explicit phylogenetic approaches are underlined.

*tet(O/W)*, *tet(Q)*, *tet(W)*, and *tetA(P)* genes. These *tet* genes have been identified in 135 genomes (29.87%) examined within this study, disseminated across 103 genomospecies (41.37%), 72 genera (52.17%), 17 families (51.51%), and 4 phyla (50%). In summary, we suggest that even though only a few number of genes/alleles are engaged in HGTs, they can be prevalent and commonly shared by different bacterial genomes.

## Advantages and limitations of the purposed pipeline, comparison with other computational tools

Finally, we employed other computational tools to identify HGT genes within our data set, aiming to assess the strengths and weaknesses of our pipeline. We first adjusted the settings in our pipeline to target genes sharing a 100% nucleotide identity over a ≥99% global length across different genera (referred to as "100id pipeline"). In addition, we employed the MetaCHIP (23), which identifies HGTs through the detection of highly similar genes (≥75% nucleotide identity, with a gene length ≥200 bp) in distantly related bacteria, much like our pipeline. All three methods yield results that encompass lists of putative HGT genes shared by different genera or higher taxa. We identified 6,545 putative HGT genes using the proposed pipeline, 4,855 using the 100id pipeline, and 9,900 using MetaCHIP to be shared across different genera.

For our comparative analysis, we also employed composition-based tools. While AlienHunter (21) solely identifies alien sequences within a genome, ShadowCaster (19) and HGTector (22) search for the origin of alien sequences by comparing them against a database of protein sequences. In all three cases, the analysis yielded lists of putative HGTs for each genome. However, limitations in time and computational resources hindered us from performing extensive analyses using ShadowCaster and HGTector on the complete data set of 452 genomes. Consequently, we focused our efforts on analyzing HGT in a single genome, specifically that of *Phocaeicola sp900066445* 1_COKtk. The summarized outcomes are presented in Table 5. Across the various tools employed, the number of detected HGT genes varied: 74 were identified using the 100id pipeline, 77 using the standard proposed pipeline, 89 using MetaCHIP, 511 using ShadowCaster, 633 using AlienHunter, and 437 using HGTector. It is evident that composition-based approaches detect a significantly higher number of HGT genes within a single genome.

In summary, detecting HGT presents considerable challenges due to factors such as genome complexity and evolution, and variations in microevolution rates among different genes—ranging from rapid changes causing higher divergence of some genes or on the contrary causing convergent evolution to gradual changes resulting in sequence conservation across non-related bacterial taxa. Other complications include incomplete databases, sampling bias, and more. Considering all these complexities, it is understandable that there exists a lack of uniformity among various computational methods utilized to detect HGT (22).

In the following section, we will focus on conducting a comprehensive evaluation of the proposed pipeline. It is important to address some initial limitations. First, it is

crucial to note that HGT is not restricted solely to homologous sequences. For instance, flanking regions associated with IS elements or transposons can function as carriers, facilitating the transfer of genes that might lack homology or similarity with the recipient genome. Second, genes shorter than 300 bp, including those encoding recombination directionality factors, are recognized as HGT genes (48).

To simplify the detection of HGT genes, we searched for genes larger than 300 bp that are nearly identical and co-shared by different taxonomic groups. This approach is similar to settings applied elsewhere to detect both recent and past HGT events (31, 32). Conversely, the study by Groussin et al. (28) asserted that a 99% nucleotide diversity corresponds to an event timescale of approximately 0–10,000 years ago, with an average estimation of one single nucleotide polymorphism per genome per year. Such occurrences are plausible under neutral evolution. However, our unpublished data from challenge experiments involving *Salmonella* Enteritidis SE147 in a chicken host clearly demonstrated that genomes from various *Salmonella* isolates of the challenged strain can exhibit differences of up to eight single nucleotide polymorphisms within a 2-week experiment. Therefore, we assume that genes under positive selection pressure, particularly some of those transferable through HGT, have a propensity to adapt to the codon usage of the novel host genome (31). Nevertheless, we also performed the analysis using 100% average identity settings. In total, 3,676 identical HGT genes were shared across different genera, compared to the 6,545 nearly identical genes identified in our proposed analysis. This decrease in numbers was anticipated. Conversely, the shift from nearly identical to truly identical variants led to the discovery of 165 novel alleles of previously identified HGT genes. Regarding the analysis of *tet* genes, 16 identical alleles/genes were identified in the HGT gene pool shared across 72 genera (compared to 14 nearly identical *tet* genes found in 72 genera). Interestingly, when focusing on the analysis of individual genomes such as *Phocaeicola sp900066445* 1_COKtk, we detected 74 vs 77 HGT genes (Table 5). We can conclude that whether genes are widely distributed across genomes or genera and are potentially subject to positive selection, we can identify them using either identical or nearly identical criteria. Considering both the analysis and *in vivo* experiments, we hypothesize that identifying nearly identical genes is the correct approach. Researchers should be mindful that the output of any computational tool is a list of putative HGT genes, which must undergo experimental validation to establish their validity.

In the following step, we established the taxonomic boundaries and groups and created the non-redundant pan-genome for each taxonomic cluster. The other three computational tools under scrutiny defined groups of organisms with varying degrees of relatedness. While MetaCHIP utilizes the GTDB, HGTector uses NCBI RefSeq microbial genomes and ShadowCaster constructs a phylogenetic shadow using a query/NCBI database. Within our pipeline, phylogenetic relatedness has been established through the construction of phylogenetic trees and whole-genome comparison. A key advantage of this pipeline is the ability to classify not yet identified and characterized genomes. The straightforward identification of genomospecies is contrasted by the bioinformatic challenge in precisely defining higher taxa. This challenge arises from various factors, including variation in genome evolution, genome plasticity, and adaptability in different branches of bacterial phylogenetic trees. Therefore, we employed the analysis of 16S rDNA sequence in our pipeline, the analysis which has the limitation in draft genome sequence era due to the inability to reconstruct intact 16S rDNA sequences. However, with the advent of long-read sequencing, these limitations can be overcome. To my knowledge, none of the evaluated pipelines examined the redundancy of the analyzed genomes, although redundant sequences can artificially inflate the apparent diversity and may lead to biased or misleading results.

In consideration of computational resources, specifically the demand for significant computational power often required in composition-based analysis (as seen in tools like ShadowCaster and HGTector), we opted for a different approach. We employed the CD-HIT similarity search method. This tool offers versatility for the user, requiring no

database formatting and outperforming the standard BLAST analysis in terms of speed. Consequently, it is well-suited for analyzing extensive genome collections effectively.

The proposed pipeline offers several advantages, including impressive speed that allows for swift analyses. Its parameter flexibility enables adjustments to tailor the detection process. Notably, this approach does not rely on specific databases, enhancing its versatility. In addition, the pipeline can be also applied in metagenomics studies and to detect HGT among metagenomic assembled genomes. However, it is important to acknowledge its limitations, such as the potential detection of false-positive results and the fact that the pipeline is not fully automated, requiring manual intervention at certain stages.

## Conclusion

The present study paves a new avenue for the detection of HGT in the collection of sequenced genomes. Based on the statistical analysis, we hypothesize that nearly identical genes co-shared between distinct genera comprise genes are capable of being transferred from genome to genome *via* HGT, the mobilome. In general, different species and strains of the same genera usually bear different cargo of mobilized genes. The present pipeline is versatile, user-friendly and results in network visualization. Importantly, the pipeline reveals new findings regarding not yet characterized genes, genes usually co-transferred with genes involved in resistance, virulence, and/or mobilization.

## MATERIALS AND METHODS

The workflow of the presented pipeline is depicted in Fig. 1. For more details about the parameter settings used to run the pipeline, see Supplementary File 1.

### Bacterial collection, whole-genome sequencing and analysis

Altogether, 452 bacterial draft genome sequences were used in the study (Table S1). The in-house bacterial culture collection comprised bacterial isolates originating from healthy chicken cecal mass ($n = 398$), and porcine feces ($n = 54$). Taxonomical assignments and nomenclature of each genome were determined using the BLASTn comparison against NCBI RefSeq16S rRNA sequence database, as of 31 March 2023 (47). Genomic DNA (gDNA) was extracted and sequenced on the Illumina platform as described previously (31, 47). In this study, additional 195 genomes were included (Table S1). Raw sequencing reads were treated and assembled by Shovill v.0.9.0 with default settings (github.com/tseemann/shovill). In the post-processing step, the contigs were scanned for the presence of polyG tracts and Illumina adapter sequences. To avoid the presence of cross-contaminant sequences, low-coverage contigs (<10% of L50 contigs mean coverage) were removed. Finally, draft genome sequences were annotated using Prokka v1.14.5 with careful option (49) and predicted proteins were functionally characterized by the Cluster of Orthologous Genes (50) using eggNOG-mapper2 v.2.1.2. software (51). In addition, proteins of general (R category) and unknown functions (S category and uncharacterized proteins) were also screened using RPS-BLAST against the Conserved Domain Database (CDD) (52, 53). To detect antibiotic resistance genes, gene sequences were also screened against the ResFinder v.4.0 database (54). In addition, taxonomy classification based on the core genome sequences was assigned by GTDB-Tk v.1.6.0 (27, 55).

Table S1 provides an overview of the sample collection composed of chicken and porcine gut microbiota. This summary includes details like the NCBI and GTDB-Tk nomenclature, sample source and host origin, collection date, and culture conditions.

## Phylogenetic reconstruction

To reconstruct phylogenetic relatedness among 452 genomes, different phylogenetic approaches were applied. The tree based on concatenated bacterial core gene sequences was inferred *via* the UBCG2 (up-to-date core genome sequences) pipeline (56). In parallel, genes coding for 16S rRNA sequences were extracted from the annotation files, aligned by Clustal Omega v.1.2.4 (57) and the phylogenetic tree was constructed under GTR+ Γ4 substitution model using an online RAxML-NG tool (58). The phylogenetic trees were visualized in iTOL v.6.7 (59).

## Definition of genomospecies, genera, and families based on the comparative analysis of genome sequences

To assign draft genomes to particular species (genomospecies), the dRep v.3.4.0 (60) tool was employed. First, draft genome sequences were divided into seven phylogenetically distant groups (Fig. 3) based on the unrooted phylogenetic tree generated through UBCG analysis (Fig. 1). Within each of these groups, the genome sequences were subjected to comparison using dRep.

Briefly, draft genome sequences were first scanned for completeness and contamination using checkM v.1.0.7 (61), then all-against-all genome sequences were compared using Mash v.1.1.1 (62), and only groups of genome sequences with Mash distance ≥0.9 were further analyzed. Finally, nucmer from the MUMmer v.3.23 package (63) was applied to calculate the average nucleotide identity (ANImf). Genomic sequences showing ANImf ≥0.95 (35) were considered to be the same genomospecies.

Next, genome sequences were assigned to particular genera and families based on the clustering of their 16S rRNA gene *via* CD-HIT v.4.8.1 (64). Clusters of genomes with nucleotide identity ≥94.5% were considered the same genus (Genera_16S), whereas clusters ≥ 92% were considered the same family (Families_16S).

Thus, each genome was assigned to Genomospecies, Genus_16S, Family_16S, bacterial family, and phylum.

## Determination of non-redundant pan-genome

All predicted genes (≥300 bp in length) of a single genomospecies were compared and clustered using CD-HIT (64) under criteria: ≥99% nucleotide identity over ≥99% global length alignment. A set of representative coding sequences of the CD-HIT clusters was considered NRPG of the genomospecies since the set represents all genes of the particular genomospecies in our collection. Similarly, we also defined the NPRG for every genus, family, and phylum within the aforementioned groups (Genera_16S, Families_16S, Families, and Phyla).

## Identification of horizontally acquired genes

In this study, we first focused on the identification of nearly identical genes that are shared by different genomospecies. All NRPGs of all genomospecies were compared by CD-HIT and genes showing ≥99% nucleotide identity over ≥99% global length alignment were identified. Since accurate *in silico* gene calling, especially the identification of start codon, cannot be generalized in bacteria (65) and horizontally acquired gene sequences are being adapted to the host codon usage (31), our criteria allow subtle changes in gene sequences and we can still detect recent horizontal events (28). In parallel, nearly identical genes shared by different genera, families, and phyla were also identified.

## Heatmap visualization and statistical analysis

The heatmaps (Fig. S2) were created in R v.4.1.3 using R package RColorBrewer (66), and its axes were visualized in the same color scheme in the tree using iTOL. Profiles of COG categories (Fig. 5) were compared to identify genes that have been likely transferred

horizontally and to discard genes that are transferred from bacteria to bacteria *via* vertical paths. Protein function and COG category have already been assigned for all genes, including putative horizontally acquired genes (Table 1).

Six different sets of genes were identified and analyzed using R packages seqinr (67), stats (68), and tidyverse (69). These sets of genes were included in the analysis: (i) all genes across all isolates (≥300 bp in length; group All); (ii) genes shared by two different genomospecies (group Genomospecies); (iii) genes shared by two different genera based on 16S rRNA analysis (group Genera_16S group); (iv) genes shared by two different families based on 16S rRNA analysis (group Families_16S); (v) genes shared by two different taxonomically defined families (group Families); and (vi) genes shared by two different taxonomically defined phyla (group Phyla). For every gene set, the absolute and relative frequency of COG categories were calculated and compared.

Friedman's test (a non-parametric statistical test) was first utilized to assess whether there were statistically significant differences in COG profiles (using either absolute or relative frequencies) among different gene sets. The comparison of COG profiles using absolute frequencies was statistically significant. Therefore, Dunn's *post hoc* test was employed to assess the specific groups that exhibited differences. The COG profile of each identified set of HGT genes was compared to the "All" group. The *P*-values were adjusted by Bonferroni's method. The *P*-values lower than 0.05 were considered statistically significant. The analysis was performed using the statistical software GraphPad Prism v.5.04 (GraphPad, Inc., San Diego, CA, USA).

## Static network analysis

The reconstructed HGT networks and their subnetworks were created and analyzed using Cytoscape v.3.9.0 (70). The networks were reconstructed using a MatReader v.2.1 (71) and analyzed as undirected using the Analysis network tool which is included in the default setting of Cytoscape. The edge transparency was set as a value of the number of genes transferred between two nodes (genera).

## Comparison with other computational tools

To assess the advantages and limitations of the proposed pipeline, we incorporated other freely available computational tools for detecting HGT. Specifically, we employed MetaCHIP v.1.10.13 (23) pipeline and HGTector2 v.2.0b3 (22), both combining similarity and incongruence within the phylogeny approach, AlienHunter v.1.7 (21) based on sequence composition, and ShadowCaster v.0.9.2 (19) based on a hybrid approach. In addition, we applied our pipeline using the criterion of 100% identity over 99% length.

Parameter settings are summarized in Supplementary File 1.

## Approval for animal experiments

No chickens or pigs have been euthanized for this study. All DNAs originated from animal cecal or fecal samples collected in the earlier studies. The handling of animals in these studies was performed in accordance with current Czech legislation (Animal Protection and Welfare Act No. 246/1992 Coll. of the Government of the Czech Republic) and the specific experiments were approved by the Ethics Committee of the Veterinary Research Institute followed by the Committee for Animal Welfare of the Ministry of Agriculture of the Czech Republic (permit number MZe1922).

## ACKNOWLEDGMENTS

This work has been funded by the Czech Science Foundation (22-16786S) to D.C. and the Internal Grant Agency of the University of Veterinary Sciences Brno (206/2023/FVHE) to M.D. Computational resources were provided by the e-INFRA CZ project (ID:90140), supported by the Ministry of Education, Youth and Sports of the Czech Republic. The funding sources have no role in the design or carrying out of the study, data collection,

analysis or interpretation, the writing of the manuscript, or the decision to submit it for publication.

J.S., D.C., and I.R. designed the pipeline. J.S., D.C., and M.Z. implemented the pipeline and tested other tools. K.J. and M.N. contributed by 16S DNA analysis. D.C. and M.N. uploaded data to NCBI. J.S. and V.B. performed the statistical part. J.S. and M.V. made a visualization of results and network analysis. J.S. and D.C. wrote the draft manuscript. V.P., M.D., and W.W. revised the paper. All authors contributed to conceptualization, writing, review, and editing. All authors declare they have no conflict of interest.

## AUTHOR AFFILIATIONS

[1]Department of Biomedical Engineering, Faculty of Electrical Engineering and Communication, Brno University of Technology, Brno, Czech Republic

[2]Molecular Systems Biology (MOSYS), Department of Functional and Evolutionary Ecology, University of Vienna, Vienna, Austria

[3]Veterinary Research Institute, Brno, Czech Republic

[4]Department of Biology, University of Oxford, Oxford, United Kingdom

[5]Vienna Metabolomics Center (VIME), University of Vienna, Vienna, Austria

[6]Central European Institute of Technology, University of Veterinary Sciences Brno, Brno, Czech Republic

[7]Department of Biology and Wildlife Diseases, Faculty of Veterinary Hygiene and Ecology, University of Veterinary Sciences Brno, Brno, Czech Republic

[8]Department of Clinical Microbiology and Immunology, Institute of Laboratory Medicine, The University Hospital Brno, Brno, Czech Republic

[9]Biomedical Center, Faculty of Medicine, Charles University, Pilsen, Czech Republic

[10]Department of Physiology, Faculty of Medicine, Masaryk University, Brno, Czech Republic

## AUTHOR ORCIDs

Darina Cejkova ⓘ http://orcid.org/0000-0002-6989-6330

## FUNDING

| Funder | Grant(s) | Author(s) |
|---|---|---|
| Czech Science Foundation | GA22-16786S | Darina Cejkova |
| Internal Grant Agency of the University of Veterinary Sciences Brno | 206/2023/FVHE | Monika Dolejska |

## AUTHOR CONTRIBUTIONS

Jana Schwarzerova, Conceptualization, Formal analysis, Methodology, Software, Visualization, Writing – original draft | Michal Zeman, Formal analysis, Writing – review and editing | Vladimir Babak, Formal analysis, Writing – review and editing | Katerina Jureckova, Formal analysis, Writing – review and editing | Marketa Nykrynova, Formal analysis, Writing – review and editing | Margaret Varga, Visualization, Writing – review and editing | Wolfram Weckwerth, Supervision, Writing – review and editing | Monika Dolejska, Funding acquisition, Writing – review and editing | Valentine Provaznik, Supervision, Writing – review and editing | Ivan Rychlik, Conceptualization, Data curation, Methodology, Writing – review and editing | Darina Cejkova, Conceptualization, Formal analysis, Funding acquisition, Methodology, Supervision, Writing – original draft, Writing – review and editing

## DATA AVAILABILITY

Draft genome sequences of investigated genomes and their corresponding raw sequencing data are available under NCBI projects PRJNA377666 and PRJNA658263.

## ADDITIONAL FILES

The following material is available online.

### Supplemental Material

**Supplementary file S1 (Spectrum01964-23-s0001.docx).** Parameter settings on the detection of HGT events using different computational tools.

**Figure S1 (Spectrum01964-23-s0002.eps).** Phylogenetic tree inferred by the RAxML using gene sequences coding for 16S rDNA genes.

**Figure S2 (Spectrum01964-23-s0003.svg).** Detailed resolution heatmap showing the abundance of genes co-shared by two different genomospecies backtracked to individual genomes with the emphasis on the visualization of low-abundant HGT genes.

**Figure S3 (Spectrum01964-23-s0004.svg).** Network visualization of genes co-shared by different isolates and genomospecies with the emphasis on the origin of isolates. Genomes of isolates presented as nodes; the same genomospecies are clustered into circle together. Edges represents the number of mobile elements transferred using edge transparency (none or white color represents zero co-transferred genes, black represents the maximum number of transferred genes, it was 548 between SAMN15872594 and SAMN15872602). Porcine isolates are marked in pink, whereas chicken isolates are in yellow.

**List of Supplementary Material (Spectrum01964-23-s0005.docx).** Legends for all supplementary files.

**Table S1 (Spectrum01964-23-s0006.xlsx).** List of analyzed samples.

**Table S2 (Spectrum01964-23-s0007.xlsx).** List of identified HGT genes.

### Open Peer Review

**PEER REVIEW HISTORY (review-history.pdf).** An accounting of the reviewer comments and feedback.

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
