## [Reviewer comments · Microbiology Spectrum]

Microbiology Spectrum

Detecting Horizontal Gene Transfer among Microbiota: An Innovative Pipeline for Identifying Co-Shared Genes within the Mobilome through Advanced Comparative Analysis

Jana Schwarzerova, Michal Zeman, Vladimir Babak, Katerina Jureckova, Marketa Nykrynova, Margaret Varga, Wolfram Weckwerth, Monika Dolejska, Valentine Provaznik, Ivan Rychlik, and Darina Cejkova

Corresponding Author(s): Darina Cejkova, Vysoke uceni technicke v Brne

Review Timeline:

Submission Date:	May 9, 2023
Editorial Decision:	June 15, 2023
Revision Received:	September 2, 2023
Editorial Decision:	September 22, 2023
Revision Received:	October 16, 2023
Accepted:	October 31, 2023

Editor: Feng Gao

Reviewer(s): Disclosure of reviewer identity is with reference to reviewer comments included in decision letter(s). The following individuals involved in review of your submission have agreed to reveal their identity: Johannes Wöstemeyer (Reviewer #1); Shay Tal (Reviewer #2); Alejandro Piña-Iturbe (Reviewer #4); Andrew S Lang (Reviewer #10)

Transaction Report:

DOI: <https://doi.org/10.1128/spectrum.01964-23>

June 15, 2023

Dr. Darina Cejkova
Vysoke uceni technicke v Brne
Department of Biomedical Engineering
Technicka 12
Brno 61600
Czech Republic

Re: Spectrum01964-23 (Detecting Horizontal Gene Transfer: A Novel Methodology Pipeline for Identifying Co-Shared Genes and Mobilome Activity based on Homology Search)

Dear Dr. Darina Cejkova:

Link Not Available

Sincerely,

Feng Gao

Journals Department
Reviewer comments:

Reviewer #1 (Comments for the Author):

Jana Schwarzerova et alii present a contribution dealing with an important problem of high interest for major parts of the community. Horizontal gene transfer is indeed of major interest for those thinking about spreading of genes across species, genus or family borders and more, in pro- as well as in eukaryotes.

The reviewer is especially attracted by the point that the authors' approaches can be used for many different organismic groups. These studies are also important for readers working on bacterial groups other than those being addressed here. This point

makes it especially important to use very clear wording, facilitating understanding of the publication. The authors are asked to consider this point in more detail.

The tremendous wealth of data - after adding the sequences to the usual data bases - makes this contribution valuable to the community. But, what is more, the provide generally applicable tools for recognising potential horizontal gene transfer. The following remarks are meant to improve readability and thus recognition of the publication.

Title: The reviewer proposes to make clear already here that the data are derived from gut bacteria. Very early in the manuscript, it must also be clear that the isolates come exclusively from porc and chicken.

Sometimes the authors use words that are scientifically more complicated than they sound. 'Homology' means a lot more than sequence similarity or identity. The authors should reconsider, if they really need the terminus homology. Another word that might provoke unnecessary criticism is 'methodology'. Literally it means the science of methods, This is not what the authors want to say. It is complicated enough to distinguish between techniques and methods, which are not synonymous.

There are reasons other than horizontal gene transfer, being responsible for nearly identical sequences in different species. The authors should provide a short discussion of such mechanisms.

The authors seem to imply that horizontal gene transfer is strictly limited to high similarity regions. This is not necessarily true. Flanking regions made of IS-elements or transposons or shorter sequence identities could easily mediate gene transfer followed by recombination in the recipient. These points need additional comments and discussion.

Line 53 also needs a bit more attention. The authors say "The acquisition of genes mediated by HGT enables bacteria to adapt to ever-changing environmental conditions." This is true, but the impression should be avoided that horizontal gene transfer is regarded as the only mechanism in this direction.

The reviewer misses information about experimental approaches to study horizontal gene transfer in the organismic groups mentioned. There is not too much published, but the relevant literature should shortly be mentioned. The reader would be more convinced to trust in the novel approach to identify horizontal gene transfer, if it has been proven by laboratory experiments before.

The author's data should also be seen as a help for those colleagues going in the experimental treatment of horizontal gene transfer.

Could the authors provide - at least in some cases - information about the endpoints of recombination? Are there IS-elements or transposons involved? Even if not, a corresponding discussion would be very helpful.

There are seemingly very few language mistakes. The reviewer saw only one in line 167. The authors are asked to recheck the text.

Treatment with some formal points could easily increase readability of the text.

It is recommended to use as few abbreviations as possible, as these always interrupt fluent reading. By no means abbreviations should be used before explanation, as seems to be the case with MGE (lines 95 and 101).

Line 186: "The species of Bacteroidaceae family ... ". The ending aceae already identifies the bacterial group as a family. Maybe doubling information is not necessary here.

The referee strongly recommends to improve figure and table captions. Many readers focus mainly or exclusively on figures and tables. Thus, the captions should provide all necessary information without reference to the main text. It is also dangerous to use abbreviations here.

Some of the figures are hard to read due to low contrast. Please also consider that publications are often printed in b/w as pdf. All information must clearly be transported under these conditions. Please recheck the use of adequate colours, improvement of contrast, size and resolution.

Reviewer #10 (Comments for the Author):

The manuscript by Schwarzerova and colleagues analyzes the genomes of >400 isolated bacteria from chickens and pigs with the goal of identifying genes moved among them by HGT. This is both interesting and important in the contexts of genome evolution and the movements of virulence and antibiotic resistance genes. To accomplish their goal, the authors have developed a new pipeline for identification of genes moved by HGT.

Some comments:

the writing/English needs improvement

As mentioned, this is an interesting and important topic, and the analyses appear to have been done carefully. However, I found the level at which the findings are presented to be at such a high level that it was difficult to assess their specific impacts. For example, there is a network analysis performed to show the connections among genes in different genomes, but the specific important findings from this analysis are not clear. Perhaps reducing some of the text on the general findings and replacing this with some concrete meaningful examples that could include genome contexts for genes moved among organisms by HGT. Do these also further support the homology-based conclusions for HGT events?

the term "co-shared" is redundant - if the genes are shared then they are present in both genomes

sp. and spp. should be plain text and not italics

line 94 "suspicious suspected of for" needs edited

line 100 wording is a bit confusing - are the gut microbiota densely inhabited or the animals?

line 109-110 - is there a figure that could be referred to here?

line 131 - statistical analysis

line 151 - I think "phylogenetically distinct" is meant? (also figure 3 legend)

line 186 "main contributors shared genes" needs edited

line 235 - I assume amino acid is meant

line 265+ text seems to be very repetitive with line 186+

line 336-338 conclusion text - I struggle with this statement - if the genes were identified as having moved by HGT, then they would indeed be capable of being transferred? is there a reason some genes can be transferred and others not? similarly, is the next sentence a conclusion based on data in the manuscript? I don't recall seeing anything so specific in the results.

figure 5 legend - would be helpful to include the statistical test performed to generate the given p-values

figure 6 - the lines between nodes are difficult to see so suggest they be made darker

the authors might want to discuss some of their findings in the context of previous work by Gogarten and colleagues that identifies biases in HGT events, specifically [10.1073/pnas.1001418107](https://doi.org/10.1073/pnas.1001418107) and [10.1038/nrmicro2593](https://doi.org/10.1038/nrmicro2593)

Reviewer #12 (Comments for the Author):

The authors present a pipeline for identifying HGT events across bacterial genera. Initially I was very excited about the paper, but there are a few questions that came up for me when reading. First, the authors talk about the importance of using culture-independent methods to detect HGT, however use cultured isolates; so can this method be used in a culture independent experiment? Second, as written the text makes the pipeline sounds specific to this dataset, can it be applied more broadly? Third, is there a website that hosts the code for this pipeline? If so what is the link? The manuscript in general needs to be proofread for clarity, and the methods need to be described in the results section in brief (for example, which types of phylogenetic trees were constructed, and which statistical tests were used). Finally, I think the authors should describe other methods for detecting HGT (eg. GUBBINS) and compare these established methods to their own.

LINE 40: What statistical analysis was used? Describe here or omit this sentence.

Line 54: Merge the two sentences on this line, they are related to one another and as written now read as two separate ideas.

Line 66: HGT is a form of recombination in some species. Be careful with the phrasing here.

Line 72-73: The text here is unclear. Please re-write.

Line 76-77: I do not think that you can limit studies on HGT to a single genera, many people are interested in detecting HGT for different applications across genera.

Line 78-80: These sentences are unclear and need to be revisited. For example "MGEs and their ability to be further dispersed." Seems to be out of place, please integrate into the prior or following sentence.

Line 82-83: Describe what massive antibiotic usage means. All farming? 50% of farms, is there a context you can put this in.

Also resistance in gut microbiota is likely also impacted by antibiotic consumption in the human population, not just in relation to farming, please describe here.

Line 84: On line 84 the authors talk about the importance of using culture-independent methods to detect HGT, however on line 91 they say that their method has only been used on cultivated isolates. Please resolve this discrepancy.

Line 103: There is missing text here.

Line 108: What phylogenetic analysis? Describe your methods. WGS? A single gene?

Line 130-132: Please describe the statistical analysis conducted.

Line 146: what is the limitation due to increasing phylogenetic distance. Describe.

Line 148: Define what a genomospecies is.

Line 169: I think the listed genomospecies would be better represented as a table, rather than a numbered list in text.

Line 186: How do you determine directionality of transfer of HGT genes?

Line 261: Merge this paragraph with the previous or following. It is too short to stand on its own.

Line 335: Is this pipeline specific to the isolates in your collection, or a generalized method that others could apply? This sentence makes it sound specific to this dataset only. Please clarify.

Line 339: Is there a place to download this pipeline for others?

Reviewer #2 (Comments for the Author):

In this manuscript, the authors describe a pipeline for detecting potential horizontal gene transfer (HGT) among bacterial genomes based on the presence of nearly identical genes co-shared between different taxonomic units. The pipeline involves the identification of coding regions in the genomes, construction of a phylogenetic network, identification of the nearly identical co-shared genes, characterization of their function, and visualization through network and heatmap analysis. The authors demonstrate the use of this pipeline for investigating the genomes of 452 isolates of chicken or porcine origin previously collected by the authors.

Overall, I find the manuscript to be interesting and scientifically valid. However, I have the following comments and concerns:

1. The authors describe the pipeline as novel in the title, abstract and text. However, the use of co-shared genes as an indication for HGT is not new, as some examples are cited in the manuscript (e.g., references 17 and 18). While the authors may have used different parameters and specific tools, the pipeline itself is not novel. I would recommend avoiding the use of the word "novel".
2. The authors used the criteria of "{greater than or equal to} 99% nucleotide identity over {greater than or equal to} 99% global length alignment" and a minimum length of 300 bp to identify potential HGT. However, according to Groussin et al. 2021 (ref 17), 99% identity over 500 bp may relate to events of HGT from 10,000 years ago. While it is acceptable to use less strict criteria (as explained by the authors in the text), the time scale should be discussed, especially since the authors mention "long-term application of antibiotics in intensive agriculture" and "commensal farm-animal".
It would be interesting to know if there are any instances of HGT with 100% identity among the results and to discuss this subgroup of HGT separately. For example, do the resistance genes come from more recent HGT events?
3. The description of the bacterial draft genomes used in this study is not clear. In the Methods section the authors state that they used 398 isolates from healthy chicken cecal mass (line 341) and 54 isolates from porcine feces (line 342). However, Table S1 (a list of all isolates used in this study) lists 392 chicken isolates and 60 porcine isolates. The authors refer to their previous studies (19, 33), but these references only describe chicken isolates. Additionally, in line 344 they mention the inclusion of "additional 173 genomes" without specifying what these genomes are and how they were collected. The source of the samples might be relevant, especially since the authors claim that the "animals were reared in commercial as well as backyard farms, were of different breed, age and sex, and were fed with food supplemented with probiotics" (lines 137-138), while the references relate to a single sampling. Moreover, since they use cultivated isolates, the choice of cultivation media may affect the outcome.
4. In lines 319-320, the authors claim that "only minority (31/6545) of genes suspected to HGT were involved in the resistance mechanism". However, previous studies (e.g., references 17 and 26), including the authors' previous studies (e.g., ref 19), indicate a much higher prevalence of antibiotic resistance genes in the mobilome. Moreover, immediately after, in the paragraph starting in line 321, the authors claim that their findings are consistent with previous studies and that the resistance genes were detected and enriched. This paragraph is unclear, I believe the findings regarding the resistance genes should be further discussed and clarified. If indeed the resistance genes are a minority, this is a very interesting finding. Could it be attributed to the less strict criteria used which may lead to the detection of HGT events prior to the extensive use of antibiotics?
5. In lines 83-88, the authors describe the advantages of using culture-independent techniques, but then they use genome sequences of cultivated bacteria for their study. I think there should be an explanation about the advantages of studying cultured systems.
6. The authors emphasize the finding of genes with unknown functions as one of the highlights of their findings. However, such genes are very common, and according to Figure 5, they are actually underrepresented in the co-shared genes compared to their abundance in the entire genomes of the isolates.
7. In Figure 1, panel III] it says "identification of nearly identical protein sequences" while the nucleic acids sequences (and not proteins sequences) were used in the pipeline. Please correct.
8. In Figure 6, the authors should add a sentence to the figure legend explaining what the nodes and edges represent.
9. In line 224 it says "888 genes were co-shared across 8 phyla", whereas Table 1 shows that there are 926 co-shared genes across 8 phyla (and 888 of them are known genes).

10. The paragraph starting in line 321 is not clear.

11. In general, I feel that the manuscript would benefit from professional editing, as some parts of it are not coherent and are hard to understand.

Reviewer #4 (Comments for the Author):

In this manuscript, the authors describe the use of a bioinformatic analysis pipeline aiming to identify horizontally transferred genes among a collection of genome sequences. The identification of HGT-genes is based on: i) identification of shared genes with high identity and coverage among all genomes, genomospecies, and within higher order taxa, ii) statistical comparison of COG profiles present in the shared gene pool of a determined taxonomic level VS the "all" gene pool, and iii) analysis of the highly similar genes shared by taxa showing an statistically different COG profile from the "All" COG profile. They apply this pipeline on 452 draft genomes obtained from bacterial isolates that belong to pig and chicken intestinal microbiota.

Major comments:

1. It is not clear whether the manuscript is oriented to present the authors' findings about HGT in the pig and chicken microbiota or the utility of their pipeline. The title seems to indicate the second option. However, throughout the manuscript, it is not clear how the applied pipeline compares to other previously published pipelines/software in terms of advantages, limitations, and major applications (Which identity thresholds should be applied? How different identity thresholds affect the results? Are there HGT-genes that are overlooked by the proposed pipeline? Which were those genes?). On the other hand, the finding that nearly identical genes shared by different bacterial genera likely participated in HGT is not a new observation. Please, include a discussion about your findings in terms of the antimicrobial-resistance genes found.
2. What are the findings if the genes shared at the Family level are analyzed? Do the COG categories change or remain the same? Is there an enrichment of a particular COG category likely involved in HGT?
3. The Methods Section needs to be rewritten to be more friendly with the reader in order to provide a guide to follow the proposed pipeline. See an example here: <https://www.frontiersin.org/articles/10.3389/fenvs.2022.901917/full>.
4. Is it possible to provide the command line used by the authors to apply this pipeline? This would provide to be highly helpful for the researchers trying to apply this pipeline.
5. In my opinion, contrary to what is stated in the Conclusions, this pipeline is not as intuitive and has many limitations. However, it seems to be very helpful to provide a preliminary set of shared genes that can be further characterized or analyzed in order to assess their involvement in HGT. I believe the conclusions should be centered on this.
6. While the title states that the pipeline identify mobilome activity, the results only show the identification of highly identical shared genes annotated as encoding proteins involved in mobilization. In my opinion this cannot be stated as detection of "mobilome activity".
7. The introduction needs to be restructured to better address the importance of the detection of HGT in the animal gut microbiota, the current detection pipelines/approaches and the necessity of a pipeline that can do what the authors propose.
8. Lines 103-112. Is there missing text here? What were those 8 phyla? Additionally, the findings described in lines 108-112 are not shown in the phylogenetic trees provided, which are colored only by Family.
9. Figures require adjustments for a more detailed presentation.
 - a. Fig 2. Require the identification of the bacterial phyla. Brackets or an additional color ring (since iTOL was used) could be added.
 - b. Fig. 3. Please modify the figure to better convey the information. How were the genomospecies grouped in the 7 phyla? How this "sub-grouping improved and fastened the comparison"?
 - c. Fig.4. Shared genes between the same or closely related species dominate the heatmap. The genes shared by more distantly related species are difficult to see. Please, correct this. You could use a different color instead of white in the "number of genes" scale that allows you to highlight the shared genes that are currently showed in very light blue color.
 - d. Fig. 5. Please clearly indicate what are the comparisons (which group VS which group, or the subject of the comparison) that resulted in a $p < 0.05$. You could use brackets or lines. Figure 5 could be used to explain how the comparison of COG profiles results in identification of the HGT-genes since this is not clear in the main text.
 - e. Fig. 6. Each color represents a Family, however there are many similar colors, diffculting the visual identification. Please, write the names of important families or Families involved in important findings, near their location in the corresponding networks.
10. In Table 1. Is not clear whether the Identified genes suspicious of HGT are from the non-redundant pool (NRPG) or from the 1,235,343 pool. Also, it would be helpful to provide the numbers of the NRPG genes identified. Also, are the "unknown genes" genes of unknown function? Please clarify or correct.
11. Lines 219-221. This explanation of 16S identity thresholds belong to the Methods section.
12. Methods Section, line 344. The procedence of the bacterial culture collection is unclear. Please detail (or cite and summarize) the procedure of isolation of the bacterial strains. Also, it is unclear if the additional 173 genomes are part of the 452 or not. Please clarify.
13. Line 279. "We consider co-shared genes between different genera are very likely to be mobilized." It is not clear how the previously presented reasoning supports this conclusion. Please, clarify.
14. Methods Section, lines 418-422. It is not clear what is the Dunn's test comparing. Also in Fig. 5. Is it a given COG profile VS the "All" category? Or VS any other category? Moreover, the text does not explain how comparison of COG categories results in identification of HGT genes which is the main focus of the paper. Based on what is written in the Results/Discussion section in

Line 239-244, it seems to be a comparison between the COG profile of a given group VS the "All" group. However, this is not understood from Fig. 5 and is not described in the Methods Section.

15. Related to the previous comment. How is the statistical analysis carried out? I'm not sure how a categorical COG profile can be compared to another with the Friedman and Dunn tests?

Minor comments:

Line 34: Is it necessary to write "co-shared"? "Shared" already conveys the idea of all the involved subjects (genomes) having something (gene) in common.

Line 46: "only several genes were co-shared between.." Please rephrase to specify how many genes.

Lines 66-68: Does HGT is the consequence or the cause of the adaptation/competition capacity of microorganisms? The cited reference does not deal with HGT. Please cite the correct reference supporting the claim.

Line 77: How does the conventional methods for HGT detection rely on ... the analysis among multidrug resistant pathogens? The meaning of this sentence is unclear.

Lines 78-80: the phrase in these lines seems to be more related to the previous paragraph than the paragraph in which is currently placed.

Line 84: Here the utility of culture-independent techniques is presented. However, below it is stated that the work was carried out with isolated bacteria. Please remove the phrase about culture-independent techniques.

Line 94: "genes suspicious suspected of for HGT". Please correct.

Line 117: Please reference the corresponding figure showing what was described in the text.

Line 120: "To identify HGT genes" instead of "traits".

Line 131: Please, indicate the statistical analysis used.

Line 145: Please, correct the tense of the text, since the past tense here gives the notion that you are describing your findings. However, for the cited reference, you are actually presenting a previous published observation.

Line 150: Please, reference the corresponding figure showing what is described in the text.

Line 265: The sentence "In total, 6,545 unique genes were co-shared by at least two genera" can be understood as there are two genera that share 6545 genes. Is this correct? Please, clarify or correct.

Line 377: how does the evolutionary relationship was estimated before using dRep?

Line 388: why a 300 bp threshold was selected? Please, indicate the rationale and include a short discussion (in the discussion section) of how the threshold affects the results. For example, some small genes involved in HGT, such as recombination directionality factors, have ≈ 200 bp in length.

Lines 406-408. Please, specify how the comparison of COG profiles was used to identify HGT genes.

Line 431. "No chickens have been sacrificed solely for the purpose of this study." Does the "solely" word means that chickens were in fact euthanized but not only for this research project? Please, clarify. Also, assess whether "euthanized" is a more proper word instead of "sacrificed".

Reviewer #6 (Comments for the Author):

The manuscript by Schwarzerova et al. introduces a potentially valuable bioinformatic pipeline for detecting laterally transferred genes (LGT) between bacteria. The authors claim that by searching for nearly identical gene sequences in phylogenetically unrelated genomes, they can identify recent cases of LGT and shed light on the emergence of antibiotic resistance. While the research topic is promising, the current version of the manuscript falls short of conveying the true value of the data and requires further attention from the authors.

To strengthen their work, the authors should compare their novel method with existing approaches and analyze a new dataset. It is crucial for the authors to demonstrate the advantages and limitations of their method in comparison to other techniques and showcase what can be uncovered using their pipeline. They could start by applying several existing LGT detection methods to a well-analyzed set of bacterial genomes, followed by implementing their own method. Each identified case should undergo verification through multiple independent statistical techniques, and the significance measures should be compared. Only after this thorough analysis, the authors should introduce their dataset derived from chickens and pigs.

Regarding the pipeline itself, the verification of the detected cases lacks persuasiveness. While the authors employed COG groupings, additional tests are required to ensure the reliability of the results. Furthermore, the manuscript lacks information on how the authors assigned taxonomic names to the 400+ isolates. It is essential for the authors to clarify this aspect. Additionally, reference 33 presents a smaller set of isolates, which raises questions about the identification of bacteria used in the study.

A significant drawback of the manuscript is the combined presentation of Results and Discussions. This approach hinders the reader's ability to extract the true value of the work. It can be acceptable for some descriptive studies but not for a bioinformatic pipeline. The authors should clearly distinguish between these sections and provide a concise summary of their findings. As it stands, it is unclear whether the manuscript primarily focuses on the development of a novel pipeline or the analysis of the mobilomes of gut bacteria. Neither of these topics is adequately described, leading to an overall lack of clarity and limited incremental value of the manuscript.

Minor: the ms text needs revision regarding overstatements and the use of terms

Reviewer #7 (Comments for the Author):

The manuscript "Detecting Horizontal Gene Transfer: A Novel Methodology Pipeline for Identifying Co-Shared Genes and Mobilome Activity based on Homology Search" describes a methodology pipeline suitable for predicting shared genes with the potential for horizontal gene transfer. The analyzed data set includes 452 genomes of bacterial isolates from the gut microbiota of animals (*Gallus gallus* and *Sus scrofa*). This manuscript addresses the current topic of HGT in the gut microbiota of animals. The study and prediction of possible gene transfer within animal breeding are topical due to the increasing numbers of resistant strains in animal breeding, with HGT being the major driver.

It is to be considered whether the manuscript should be intended as a research article and describe and discuss the actual results of the work in more depth and detail than the described methodology pipeline, which uses previously published bioinformatics tools.

If the manuscript remains focused on describing the actual methodology pipeline, It would be appropriate to state in the manuscript:

1. The source code of the methodology pipeline, if available?

2. How is the proposed methodology pipeline different from previously published ones?

<https://www.science.org/doi/full/10.1126/sciadv.abj5056>

<https://doi.org/10.1186/s40168-019-0649-y>

<https://doi.org/10.3390/genes11070756>

3. What are the weaknesses of the design analysis approach?

I have the following other comments and recommendations that I would like the authors to address:

line 64: This sentence is not related to the topic.

line 102 - 109: There are missing eight phyla mentioned in the text; the taxonomy given in the whole text does not respect the currently valid taxonomy of prokaryotes, please modify according to <https://doi.org/10.1099/ijsem.0.005056>

line 107: Please clarify, what are the 81 conserved genes, on the basis of what criteria were they selected?

line 111: Fig S1 - It is not clear what is the main message of this figure. It would be useful to highlight the families belonging to each phylum mentioned in the text.

line 152: Figure 3 - I suggest moving Figure 3 to the supplementary material.

line 167 - 183: It is not necessary to list all 18 genomospecies, which do not co-shared any gene, in the text, if this information is shown in Figure 4.

line 291: Table 2 - Move the table to the supplementary; this data presentation format is not suitable for the main text. Choose another form of presentation of horizontally transferred genes. Most of the identified genes are carried on mobile genetic elements. It would be interesting to categorize the genes based on which MGEs they are carried on.

line 292 - 302: Please simplify the text. The information in the main text is duplicated in the legend of Figure 6. In the main text, give the names of individual clusters of orthologous groups (COGs), not their numerical designations, which are incomprehensible to the reader. COG IDs are listed in Table 2.

In general, the main text is difficult to understand. This is due to the fact that the text combines a description of the results and a description of the methodology pipeline. I propose to move the data analysis workflow to the Material and Methodology section and instead focus on the presentation of the results in the results section.

The results should be discussed in the broader context of microbiome research.

Overall, I propose a major revision of the text.

Reviewer #8 (Comments for the Author):

In this work, the authors utilize sequencing data from over 450 isolates across 8 phyla of bacteria isolated from chicken and porcine fecal samples to examine shared gene content. The overall approach is commendable, and the pipeline has the potential to be a useful tool for the research community. However, the authors have provided only limited application of their method and neglected to include biological interpretations of their findings which undermines the work. In addition, there are significant english language edits that need to be addressed throughout the paper. Due to the extent of the revisions required, I have only included line edits below for major concerns.

There is no reason to introduce the term 'genomospecies' throughout the paper. Indicating what level of ANI (or other metric) you utilized to delineate a species and supporting that decision (as you have done through your references) is appropriate and avoids confusion.

Line 34: the study does not propose that 'nearly identical genes co-shared between distinct genera can be mobilized from genome to genome via HGT.' This sentence suggests that the first requirement is for genes to have high identity and this allows HGT to occur. Your study is proposing that you can provide evidence of recent HGT events between distinct genera by identifying these nearly identical genes.

Line 42: You speak of reservoirs of resistance, but have not indicated what these are reservoirs for? Did you find evidence that these organisms are providing AMR genes to pathogens (which I presume are the beneficiaries of these reservoirs?)

Line 77 - this is an example, not a method.

Lines 87-89 are irrelevant to your study.

Lines 140-142 What evidence do you have to support that 99% ID allows for past events?

Lines 169-185: Why is there a numbered list inserted in your paper? Include this as a proper table or discuss the major findings and move these details to the supplemental.

Lines 199-202: I'm not sure your data can support this statement. Although it is possible that bacteria are moving rapidly between different areas, it is also possible that the genes detected are under strong selection pressure to preserve their sequence identity or that similar selection pressures in different environments are promoting their acquisition from local reservoirs.

Line 204-210: Does this refer to reciprocal sharing between these species or the fraction of putatively shared genes per genome?

Lines 296-305: These are obvious choices for network analysis, but was there a systematic method to choosing genes or was it arbitrary?

Lines 317-325: your analysis indicated that AMR genes (but not the mobile elements moving them) are highly identical in Gram-positive and Gram-negative, but this suggests there is a biological explanation related to AMR that is separate from recent HGT acquisition. If these genes were identical because they had recently been acquired then you should also see evidence of their acquisition (transposons, IS elements etc.) that were also highly identical, or provide literature to support rapid loss of these MGEs.

Lines 340-341: "Importantly, the pipeline reveals new findings regarding not yet characterized genes, genes usually co-transferred with genes involved in resistance, virulence and/or mobilome activity." I don't doubt that your pipeline is capable of this, and I agree that it is a very useful tool for the research community. But you have not taken the time in this paper to show the application of this pipeline to these questions. Which hypothetical genes did you identify that should be investigated further? What evidence did you generate for the mechanism of AMR gene movement between diverse species? You identify some species with no shared genes and some that you consider reservoirs but do not discuss the biological implications of these findings or give any insight into possible reasons. You have identified genomes with co-shared genes but have not included any investigation of the nature of these co-shared genes or how this information can be used in future studies.

Reviewer #9 (Comments for the Author):

Review Schwarzerova et al - Detecting HGT - novel methodology pipeline - June 2023

This paper proposes a new in silico methodology to detect HGT among complex population, through NGS and selection of co-shared genes between "genomespecies".

I believe this method is a great interest for the community. However it is important to note that the threshold used here (>99% of nt identity and >99% length alignment), even if less stringent than previous studies, is still restrictive. As such, it only includes genes detectable using this limitation, but it excludes any gene that could be initially absent in the recipient cell (for example within MGEs), or genes that could be located between two other genes which present sufficient sequence identity to support homology recombination. Thus, I believe the authors should indicate that their study is not exhaustive at all to detect HGT, as soon as within the abstract.

Another surprising point for me is the indication that this pipeline is "intuitive, easy to use" (l. 339). I believe if you are not trained in bioinformatics, it is not easy to use at all. To the best of my reading of the manuscript, the authors do not provide here a website or an online platform where each microbiologist over the world could load a set of genomes, looking for HGT between their genomes using this pipeline. This would really be easy to use. I do not mean that this would be a request as referee, however, please do not think that all microbiologists are computer scientists.

Be careful L103-107 are empty lines!!!

L120-132: in my opinion, this crucial part of the paper is not so clear for microbiologists who are not computer scientists. For example, I did not understand what is dREP? I thought it would be explained later, but it is not explained either in l.147. The paper should be easily understandable for any microbiologist.

In the same line, in Figure 1 legend, please provide the meaning of each acronym. Some are understandable (RAxML, ClustalOmega), but some are not so used to my knowledge (UBCG, iTOL, dRep, eggNOG-mapper...). Perhaps each person would be familiar with different acronyms, but not all of them. So please make the figure comprehensible by the legend.

L137 the reader learn important things about the collection used (collection over 5 years...). I believe the collection should be presented in more details in the section "Bacterial diversity of animal gut ..."

L199-202: the observation of HGT between strains from chicken gut and porcine gut is interesting. However, these HGT cannot be direct from chicken gut bacteria to porcine gut bacteria. Why the authors did not discuss the probability that one or several intermediate HGT would have occurred?

Other minor comments:

- L54: I think that the misuse of antibiotics in human medicine is also responsible for the dissemination of AMR, leading to well known nosocomial infections.
- L55-56: Gut microbiota is not the only important reservoir of AMR genes.
- L94: please rephrase
- L98: I think the authors should add "analysed in this work" at the end of the title of this section. The authors did not analysed the animal gut microbiota of all animals, nor in all countries...
- L111 why "also"? Does another phylogenetic analysis is involved here? (unless 16S rDNA)
- L129 please explain CD-HIT
- L130 please explain eggNOG-mapper
- L169-185 why the authors did not provide this list as a Table?
- L283: to assess
- L324 I guess the authors means antibiotic resistance, or any kind of resistance including to heavy metals, endonucleases and other bacteriocins?
- L335 please replace "the collection" by "our collection" as this is a specific study based on a specific collection.
- L337 comprise genes which are capable to be...
- Reference number 10 should be replaced by another one which have been peer reviewed. I do not think that such reference from Internet have been peer reviewed.
- Figure 3 legend please identify 1 to 7 in Legend
- Figure S1 the phylogeny could be enlarged
- Figure 2 and S1, it could be nice to indicate the different phyla on the left of the colored ranges.

Staff Comments:

Preparing Revision Guidelines

Please return the manuscript within 60 days; if you cannot complete the modification within this time period, please contact me. If you do not wish to modify the manuscript and prefer to submit it to another journal, please notify me of your decision immediately so that the manuscript may be formally withdrawn from consideration by Microbiology Spectrum.

Review for Spectrum01964-23: “Detecting Horizontal Gene Transfer: A Novel Methodology Pipeline for Identifying Co-Shared Genes and Mobilome Activity based on Homology Search”.

In this manuscript, the authors describe a pipeline for detecting potential horizontal gene transfer (HGT) among bacterial genomes based on the presence of nearly identical genes co-shared between different taxonomic units. The pipeline involves the identification of coding regions in the genomes, construction of a phylogenetic network, identification of the nearly identical co-shared genes, characterization of their function, and visualization through network and heatmap analysis. The authors demonstrate the use of this pipeline for investigating the genomes of 452 isolates of chicken or porcine origin previously collected by the authors.

Overall, I find the manuscript to be interesting and scientifically valid. However, I have the following comments and concerns:

1. The authors describe the pipeline as novel in the title, abstract and text. However, the use of co-shared genes as an indication for HGT is not new, as some examples are cited in the manuscript (e.g., references 17 and 18). While the authors may have used different parameters and specific tools, the pipeline itself is not novel. I would recommend avoiding the use of the word “novel”.
2. The authors used the criteria of “ $\geq 99\%$ nucleotide identity over $\geq 99\%$ global length alignment” and a minimum length of 300 bp to identify potential HGT. However, according to Groussin et al. 2021 (ref 17), 99% identity over 500 bp may relate to events of HGT from 10,000 years ago. While it is acceptable to use less strict criteria (as explained by the authors in the text), the time scale should be discussed, especially since the authors mention “long-term application of antibiotics in intensive agriculture” and “commensal farm-animal”.
It would be interesting to know if there are any instances of HGT with 100% identity among the results and to discuss this sub-group of HGT separately. For example, do the resistance genes come from more recent HGT events?
3. The description of the bacterial draft genomes used in this study is not clear. In the Methods section the authors state that they used 398 isolates from healthy chicken cecal mass (line 341) and 54 isolates from porcine feces (line 342). However, Table S1 (a list of all isolates used in this study) lists 392 chicken isolates and 60 porcine isolates. The authors refer to their previous studies (19, 33), but these references only describe chicken isolates. Additionally, in line 344 they mention the inclusion of “additional 173 genomes” without specifying what these genomes are and how they were collected. The source of the samples might be relevant, especially since the authors claim that the “animals were reared in commercial as well as backyard farms, were of different breed, age and sex, and were fed with food supplemented with probiotics” (lines 137-138), while the references relate to a single sampling. Moreover, since they use cultivated isolates, the choice of cultivation media may affect the outcome.
4. In lines 319-320, the authors claim that “only minority (31/6545) of genes suspected to HGT were involved in the resistance mechanism”. However, previous studies (e.g., references 17 and 26), including the authors’ previous studies (e.g., ref 19), indicate a much higher prevalence of antibiotic resistance genes in the mobilome. Moreover, immediately after, in the paragraph starting in line 321, the authors claim that their findings are consistent with previous studies and that the resistance genes were detected and enriched. This paragraph is unclear, I believe the findings regarding the resistance genes should be further discussed and clarified. If indeed the resistance genes are a minority, this is a very interesting finding. Could it be attributed to the less strict criteria used which may lead to the detection of HGT events prior to the extensive use of antibiotics?

5. In lines 83-88, the authors describe the advantages of using culture-independent techniques, but then they use genome sequences of cultivated bacteria for their study. I think there should be an explanation about the advantages of studying cultured systems.
6. The authors emphasize the finding of genes with unknown functions as one of the highlights of their findings. However, such genes are very common, and according to Figure 5, they are actually underrepresented in the co-shared genes compared to their abundance in the entire genomes of the isolates.
7. In Figure 1, panel III] it says "identification of nearly identical protein sequences" while the nucleic acids sequences (and not proteins sequences) were used in the pipeline. Please correct.
8. In Figure 6, the authors should add a sentence to the figure legend explaining what the nodes and edges represent.
9. In line 224 it says "888 genes were co-shared across 8 phyla", whereas Table 1 shows that there are 926 co-shared genes across 8 phyla (and 888 of them are known genes).
10. The paragraph starting in line 321 is not clear.
11. In general, I feel that the manuscript would benefit from professional editing, as some parts of it are not coherent and are hard to understand.

In this manuscript, the authors describe the use of a bioinformatic analysis pipeline aiming to identify horizontally transferred genes among a collection of genome sequences. The identification of HGT-genes is based on: i) identification of shared genes with high identity and coverage among all genomes, genomospecies, and within higher order taxa, ii) statistical comparison of COG profiles present in the shared gene pool of a determined taxonomic level VS the “all” gene pool, and iii) analysis of the highly similar genes shared by taxa showing an statistically different COG profile from the “All” COG profile. They apply this pipeline on 452 draft genomes obtained from bacterial isolates that belong to pig and chicken intestinal microbiota.

Major comments:

1. It is not clear whether the manuscript is oriented to present the authors' findings about HGT in the pig and chicken microbiota or the utility of their pipeline. The title seems to indicate the second option. However, throughout the manuscript, it is not clear how the applied pipeline compares to other previously published pipelines/software in terms of advantages, limitations, and major applications (Which identity thresholds should be applied? How different identity thresholds affect the results? Are there HGT-genes that are overlooked by the proposed pipeline? Which were those genes?). On the other hand, the finding that nearly identical genes shared by different bacterial genera likely participated in HGT is not a new observation. Please, include a discussion about your findings in terms of the antimicrobial-resistance genes found.
2. What are the findings if the genes shared at the Family level are analyzed? Does the COG categories change or remain the same? Is there an enrichment of a particular COG category likely involved in HGT?
3. The Methods Section needs to be rewritten to be more friendly with the reader in order to provide a guide to follow the proposed pipeline. See an example here: <https://www.frontiersin.org/articles/10.3389/fenvs.2022.901917/full>.
4. Is it possible to provide the command line used by the authors to apply this pipeline? This would provide to be highly helpful for the researchers trying to apply this pipeline.
5. In my opinion, contrary to what is stated in the Conclusions, this pipeline is not as intuitive and has many limitations. However, it seems to be very helpful to provide a preliminary set of shared genes that can be further characterized or analyzed in order to assess their involvement in HGT. I believe the conclusions should be centered on this.
6. While the title states that the pipeline identifies mobilome activity, the results only show the identification of highly identical shared genes annotated as encoding proteins involved in mobilization. In my opinion this cannot be stated as detection of “mobilome activity”.
7. The introduction needs to be restructured to better address the importance of the detection of HGT in the animal gut microbiota, the current detection pipelines/approaches and the necessity of a pipeline that can do what the authors propose.
8. Lines 103-112. Is there missing text here? What were those 8 phyla? Additionally, the findings described in lines 108-112 are not shown in the phylogenetic trees provided, which are colored only by Family.
9. Figures require adjustments for a more detailed presentation.

- a. Fig 2. Require the identification of the bacterial phyla. Brackets or an additional color ring (since iTOL was used) could be added.
 - b. Fig. 3. Please modify the figure to better convey the information. How were the genomospecies grouped in the 7 phyla? How this “sub-grouping improved and fastened the comparison”?
 - c. Fig.4. Shared genes between the same or closely related species dominate the heatmap. The genes shared by more distantly related species are difficult to see. Please, correct this. You could use a different color instead of white in the “number of genes” scale that allows you to highlight the shared genes that are currently showed in very light blue color.
 - d. Fig. 5. Please clearly indicate what are the comparisons (which group VS which group, or the subject of the comparison) that resulted in a $p < 0.05$. You could use brackets or lines. Figure 5 could be used to explain how the comparison of COG profiles results in identification of the HGT-genes since this is not clear in the main text.
 - e. Fig. 6. Each color represents a Family, however there are many similar colors, difficulting the visual identification. Please, write the names of important families or Families involved in important findings, near their location in the corresponding networks.
10. In Table 1. Is not clear whether the Identified genes suspicious of HGT are from the non-redundant pool (NRPG) or from the 1,235,343 pool. Also, it would be helpful to provide the numbers of the NRPG genes identified. Also, are the “unknown genes” genes of unknown function? Please clarify or correct.
 11. Lines 219-221. This explanation of 16S identity thresholds belong to the Methods section.
 12. Methods Section, line 344. The procedence of the bacterial culture collection is unclear. Please detail (or cite and summarize) the procedure of isolation of the bacterial strains. Also, it is unclear if the additional 173 genomes are part of the 452 or not. Please clarify.
 13. Line 279. “We consider co-shared genes between different genera are very likely to be mobilized.” It is not clear how the previously presented reasoning supports this conclusion. Please, clarify.
 14. Methods Section, lines 418-422. It is not clear what is the Dunn’s test comparing. Also in Fig. 5. Is it a given COG profile VS the “All” category? Or VS any other category? Moreover, the text does not explain how comparison of COG categories results in identification of HGT-genes which is the main focus of the paper. Based on what is written in the Results/Discussion section in Line 239-244, it seems to be a comparison between the COG profile of a given group VS the “All” group. However, this is not understood from Fig. 5 and is not described in the Methods Section.
 15. Related to the previous comment. How is the statistical analysis carried out? I’m not sure how a categorical COG profile can be compared to another with the Friedman and Dunn tests?

Minor comments:

Line 34: Is it necessary to write “co-shared”? “Shared” already conveys the idea of all the involved subjects (genomes) having something (gene) in common.

Line 46: “only several genes were co-shared between..” Please rephrase to specify how many genes.

Lines 66-68: Does HGT is the consequence or the cause of the adaptation/competition capacity of microorganisms? The cited reference does not deal with HGT. Please cite the correct reference supporting the claim.

Line 77: How does the conventional methods for HGT detection rely on ... the analysis among multidrug resistant pathogens? The meaning of this sentence is unclear.

Lines 78-80: the phrase in these lines seems to be more related to the previous paragraph than the paragraph in which is currently placed.

Line 84: Here the utility of culture-independent techniques is presented. However, below it is stated that the work was carried out with isolated bacteria. Please remove the phrase about culture-independent techniques.

Line 94: “genes suspicious suspected of for HGT”. Please correct.

Line 117: Please reference the corresponding figure showing what was described in the text.

Line 120: “To identify HGT genes” instead of “traits”.

Line 131: Please, indicate the statistical analysis used.

Line 145: Please, correct the tense of the text, since the past tense here gives the notion that you are describing your findings. However, for the cited reference, you are actually presenting a previous published observation.

Line 150: Please, reference the corresponding figure showing what is described in the text.

Line 265: The sentence “In total, 6,545 unique genes were co-shared by at least two genera” can be understood as there are two genera that share 6545 genes. Is this correct? Please, clarify or correct.

Line 377: how does the evolutionary relationship was estimated before using dRep?

Line 388: why a 300 bp threshold was selected? Please, indicate the rationale and include a short discussion (in the discussion section) of how the threshold affects the results. For example, some small genes involved in HGT, such as recombination directionality factors, have \approx 200 bp in length.

Lines 406-408. Please, specify how the comparison of COG profiles was used to identify HGT genes.

Line 431. “No chickens have been sacrificed solely for the purpose of this study.” Does the “solely” word means that chickens were in fact euthanized but not only for this research project? Please, clarify. Also, assess whether “euthanized” is a more proper word instead of “sacrificed”.

This paper proposes a new in silico methodology to detect HGT among complex population, through NGS and selection of co-shared genes between “genomospecies”.

I believe this method is a great interest for the community. However it is important to note that the threshold used here (>99% of nt identity and >99% length alignment), even if less stringent than previous studies, is still restrictive. As such, it only includes genes detectable using this limitation, but it excludes any gene that could be initially absent in the recipient cell (for example within MGEs), or genes that could be located between two other genes which present sufficient sequence identity to support homology recombination. Thus, I believe the authors should indicate that their study is not exhaustive at all to detect HGT, as soon as within the abstract.

Another surprising point for me is the indication that this pipeline is “intuitive, easy to use” (l. 339). I believe if you are not trained in bioinformatics, it is not easy to use at all. To the best of my reading of the manuscript, the authors do not provide here a website or an online platform where each microbiologist over the world could load a set of genomes, looking for HGT between their genomes using this pipeline. This would really be easy to use. I do not mean that this would be a request as referee, however, please do not think that all microbiologists are computer scientists.

Be careful L103-107 are empty lines!!!

L120-132: in my opinion, this crucial part of the paper is not so clear for microbiologists who are not computer scientists. For example, I did not understand what is dREP? I thought it would be explained later, but it is not explained either in l.147. The paper should be easily understandable for any microbiologist.

In the same line, in Figure 1 legend, please provide the meaning of each acronym. Some are understandable (RAxML, ClustalOmega), but some are not so used to my knowledge (UBCG, iTOL, dRep, eggNOG-mapper...). Perhaps each person would be familiar with different acronyms, but not all of them. So please make the figure comprehensible by the legend.

L137 the reader learn important things about the collection used (collection over 5 years...). I believe the collection should be presented in more details in the section “Bacterial diversity of animal gut ...”

L199-202: the observation of HGT between strains from chicken gut and porcine gut is interesting. However, these HGT cannot be direct from chicken gut bacteria to porcine gut bacteria. Why the authors did not discuss the probability that one or several intermediate HGT would have occurred?

Other minor comments:

- L54: I think that the misuse of antibiotics in human medicine is also responsible for the dissemination of AMR, leading to well known nosocomial infections.
- L55-56: Gut microbiota is not the only important reservoir of AMR genes.
- L94: please rephrase
- L98: I think the authors should add “analysed in this work” at the end of the title of this section. The authors did not analysed the animal gut microbiota of all animals, nor in all countries...
- L111 why “also”? Does another phylogenetic analysis is involved here? (unless 16S rDNA)
- L129 please explain CD-HIT
- L130 please explain eggNOG-mapper

- L169-185 why the authors did not provide this list as a Table?
- L283: to assess
- L324 I guess the authors means antibiotic resistance, or any kind of resistance including to heavy metals, endonucleases and other bacteriocins?
- L335 please replace “the collection” by “our collection” as this is a specific study based on a specific collection.
- L337 comprise genes which are capable to be...
- Reference number 10 should be replaced by another one which have been peer reviewed. I do not think that such reference from Internet have been peer reviewed.
- Figure 3 legend please identify 1 to 7 in Legend
- Figure S1 the phylogeny could be enlarged
- Figure 2 and S1, it could be nice to indicate the different phyla on the left of the colored ranges.

The authors present a pipeline for identifying HGT events across bacterial genera. Initially I was very excited about the paper, but there are a few questions that came up for me when reading. First, the authors talk about the importance of using culture-independent methods to detect HGT, however use cultured isolates; so can this method be used in a culture independent experiment? Second, as written the text makes the pipeline sounds specific to this dataset, can it be applied more broadly? Third, is there a website that hosts the code for this pipeline? If so what is the link? The manuscript in general needs to be proofread for clarity, and the methods need to be described in the results section in brief (for example, which types of phylogenetic trees were constructed, and which statistical tests where used). Finally, I think the authors should describe other methods for detecting HGT (eg. GUBBINS) and compare these established methods to their own.

LINE 40: What statistical analysis was used? Describe here or omit this sentence.

Line 54: Merge the two sentences on this line, they are related to one another and as written now read as two separate ideas.

Line 66: HGT is a form of recombination in some species. Be careful with the phrasing here.

Line 72-73: The text here is unclear. Please re-write.

Line 76-77: I do not think that you can limit studies on HGT to a single genera, many people are interested in detecting HGT for different applications across genera.

Line 78-80: These sentences are unclear and need to be revisited. For example "MGEs and their ability to be further dispersed." Seems to be out of place, please integrate into the prior or following sentence.

Line 82-83: Describe what massive antibiotic usage means. All farming? 50% of farms, is there a context you can put this in. Also resistance in gut microbiota is likely also impacted by antibiotic consumption in the human population, not just in relation to farming, please describe here.

Line 84: On line 84 the authors talk about the importance of using culture-independent methods to detect HGT, however on line 91 they say that their method has only been used on cultivated isolates. Please resolve this discrepancy.

Line 103: There is missing text here.

Line 108: What phylogenetic analysis? Describe your methods. WGS? A single gene?

Line 130-132: Please describe the statistical analysis conducted.

Line 146: what is the limitation due to increasing phylogenetic distance. Describe.

Line 148: Define what a genomospecies is.

Line 169: I think the listed genomospecies would be better represented as a table, rather than a numbered list in text.

Line 186: How do you determine directionality of transfer of HGT genes?

Line 261: Merge this paragraph with the previous or following. It is too short to stand on its own.

Line 335: Is this pipeline specific to the isolates in your collection, or a generalized method that others could apply? This sentence makes it sound specific to this dataset only. Please clarify.

Line 339: Is there a place to download this pipeline for others?

Dear editor Feng Gao,
Thank you for providing us with the opportunity to enhance our manuscript entitled “Detecting
Horizontal Gene Transfer among Microbiota: An Innovative Pipeline for Identifying Co-Shared Genes
within the Mobilome through Advanced Comparative Analysis”, Spectrum01964-23.

We greatly appreciate the valuable comments provided by the reviewers, and we have thoroughly
incorporated them into the revised manuscript. You will find all the modifications made to the original
manuscript highlighted in yellow within the "Marked Up Manuscript" file. We have also provided
detailed responses to the comments below in this letter. Our aim is to ensure that the revised manuscript
aligns with the esteemed publication standards of Microbiology Spectrum.

**Reviewer #1 (Comments for the Author):**

Jana Schwarzerova et alii present a contribution dealing with an important problem of high interest for
major parts of the community. Horizontal gene transfer is indeed of major interest for those thinking
about spreading of genes across species, genus or family borders and more, in pro- as well as in
eukaryotes.

The reviewer is especially attracted by the point that the authors' approaches can be used for many
different organismic groups. These studies are also important for readers working on bacterial groups
other than those being addressed here. This point makes it especially important to use very clear wording,
facilitating understanding of the publication. The authors are asked to consider this point in more detail.

The tremendous wealth of data - after adding the sequences to the usual data bases - makes this
contribution valuable to the community. But, what is more, the provide generally applicable tools for
recognising potential horizontal gene transfer.

The following remarks are meant to improve readability and thus recognition of the publication.

Response: Thank you for dedicating time to consider and review our manuscript. We highly appreciate the
feedback provided by Reviewer #1, which we have diligently incorporated into the revised manuscript.

Title: The reviewer proposes to make clear already here that the data are derived from gut bacteria. Very
early in the manuscript, it must also be clear that the isolates come exclusively from porc and chicken.

Response: The purposed pipeline, the subject of the manuscript, is suitable also for other collection of
microbes including metagenome assembled contigs. To our opinion, the emphasis on the gut microbiota
in the title misleading.

The first mention about porcine and chicken gut microbiota is present early in the Abstract (line 32).

Regarding the Title itself, Reviewers #1, #2, and #4 asked us to make some changes, therefore the

manuscript is entitled “Detecting Horizontal Gene Transfer among Microbiota: An Innovative Pipeline for
Identifying Co-Shared Genes within the Mobilome through Advanced Comparative Analysis.”

Sometimes the authors use words that are scientifically more complicated than they sound. 'Homology'
means a lot more than sequence similarity or identity. The authors should reconsider, if they really need
the terminus homology. Another word that might provoke unnecessary criticism is 'methodology'.
Literally it means the science of methods, This is not what the authors want to say. It is complicated
enough to distinguish between techniques and methods, which are not synonymous.

Response: we discarded the word methodology (e.g. in the Title) and replace term “homology search”
with “comparative analysis”; however we preserved the term “homologous sequences” in the text.

There are reasons other than horizontal gene transfer, being responsible for nearly identical sequences in
different species. The authors should provide a short discussion of such mechanisms.

Response: the short discussion has been added, lines 382-383.

The authors seem to imply that horizontal gene transfer is strictly limited to high similarity regions. This
is not necessarily true. Flanking regions made of IS-elements or transposons or shorter sequence identities
could easily mediate gene transfer followed by recombination in the recipient. These points need
additional comments and discussion.

Response: the short discussion has been added, lines 387-390.

Line 53 also needs a bit more attention. The authors say "The acquisition of genes mediated by HGT
enables bacteria to adapt to ever-changing environmental conditions." This is true, but the impression
should be avoided that horizontal gene transfer is regarded as the only mechanism in this direction.

Response: we slightly modified the text

The reviewer misses information about experimental approaches to study horizontal gene transfer in the
organismic groups mentioned. There is not too much published, but the relevant literature should shortly
be mentioned. The reader would be more convinced to trust in the novel approach to identify horizontal
gene transfer, if it has been proven by laboratory experiments before.

Response: This is a little tough comment. Reviewer #1 ask us to add notes about experimental
approaches, Reviewer #2 to add notes about culture-dependent techniques, Reviewer #4 to remove notes
about culture-independent techniques and Reviewers #4 and #6 to focus on current bioinformatic
techniques to detect HGT. We modified the text in l. 84-110.

The author's data should also be seen as a help for those colleagues going in the experimental treatment of
horizontal gene transfer.

Response: agreed

Could the authors provide - at least in some cases - information about the endpoints of recombination?
Are there IS-elements or transposons involved? Even if not, a corresponding discussion would be very
helpful.

Response: this has not been discussed in the ms due to length constrains.

There are seemingly very few language mistakes. The reviewer saw only one in line 167. The authors are
asked to recheck the text.

Response: corrected

Treatment with some formal points could easily increase readability of the text.

Response: Thank you for the comment. The manuscript has been reviewed by layperson in terms of
genomics as well as by native speaker,

It is recommended to use as few abbreviations as possible, as these always interrupt fluent reading. By no
means abbreviations should be used before explanation, as seems to be the case with MGE (lines 95 and
101).

Response: for the MGE abbreviation, the explanation have already been introduced in l. 70. In terms of
other abbreviations, I have found the following instances within the manuscript: NCBI, HGT, MGE, IS,
ARG, COG. To my opinion, all are commonly used in genomics and horizontal gene transfer-related
manuscripts. Other term (e.g. GTDB, dRep, MetaCHIP) are directly the names of the pipeline or
database, no abbreviation has been introduced by us. And lastly, we used term NRPG (non-redundant pan-
genome) throughout the text. However, I do understand that for novice and layman in the subject, it is
difficult to understand all term.

Line 186: "The species of Bacteroidaceae family .. ". The ending .. aceae already identifies the bacterial
group as a family. Maybe doubling information is not necessary here.

Response: agreed.

The referee strongly recommends to improve figure and table captions. Many readers focus mainly or
exclusively on figures and tables. Thus, the captions should provide all necessary information without
reference to the main text. It is also dangerous to use abbreviations here.

Response: Similar comments were also provided by Reviewer #4. Done.

Some of the figures are hard to read due to low contrast. Please also consider that publications are often
printed in b/w as pdf. All information must clearly be transported under these conditions. Please recheck
the use of adequate colours, improvement of contrast, size and resolution.

Response: Contrast, size and resolution have been increased. However, the files are than bigger than the
figure limits provided by the journal. Even in the supplementary material, the authors cannot provide
Figures in high resolution, not in .svg files.

**Reviewer #2 (Comments for the Author):**

In this manuscript, the authors describe a pipeline for detecting potential horizontal gene transfer (HGT)
among bacterial genomes based on the presence of nearly identical genes co-shared between different
taxonomic units. The pipeline involves the identification of coding regions in the genomes, construction
of a phylogenetic network, identification of the nearly identical co-shared genes, characterization of their
function, and visualization through network and heatmap analysis. The authors demonstrate the use of this
pipeline for investigating the genomes of 452 isolates of chicken or porcine origin previously collected by
the authors.

Overall, I find the manuscript to be interesting and scientifically valid. However, I have the following
comments and concerns:

Response: Thank you for dedicating time to consider and review our manuscript. We highly appreciate the
feedback provided by Reviewer #2, which we have diligently incorporated into the revised manuscript.

1. The authors describe the pipeline as novel in the title, abstract and text. However, the use of co-shared
genes as an indication for HGT is not new, as some examples are cited in the manuscript (e.g., references
17 and 18). While the authors may have used different parameters and specific tools, the pipeline itself is
not novel. I would recommend avoiding the use of the word "novel".

Response: We agree with your suggestion and replace the word “novel” with “innovative,” such as in the
title. Regarding the Title itself, Reviewers #1, #2, and #4 asked us to make some changes, therefore the
manuscript is entitled “Detecting Horizontal Gene Transfer among Microbiota: An Innovative Pipeline for
Identifying Co-Shared Genes within the Mobilome through Advanced Comparative Analysis.”

2. The authors used the criteria of "{greater than or equal to} 99% nucleotide identity over {greater than
or equal to} 99% global length alignment" and a minimum length of 300 bp to identify potential HGT.
However, according to Groussin et al. 2021 (ref 17), 99% identity over 500 bp may relate to events of
HGT from 10,000 years ago. While it is acceptable to use less strict criteria (as explained by the authors
in the text), the time scale should be discussed, especially since the authors mention "long-term
application of antibiotics in intensive agriculture" and "commensal farm-animal".

It would be interesting to know if there are any instances of HGT with 100% identity among the results
and to discuss this sub-group of HGT separately. For example, do the resistance genes come from more
recent HGT events?

Response: We totally agree with your comments. Both, the analysis and the discussion have been added,
see lines 361-379 and 394-415.

3. The description of the bacterial draft genomes used in this study is not clear. In the Methods section the
authors state that they used 398 isolates from healthy chicken cecal mass (line 341) and 54 isolates from
porcine feces (line 342). However, Table S1 (a list of all isolates used in this study) lists 392 chicken
isolates and 60 porcine isolates. The authors refer to their previous studies (19, 33), but these references
only describe chicken isolates. Additionally, in line 344 they mention the inclusion of "additional 173
genomes" without specifying what these genomes are and how they were collected. The source of the
samples might be relevant, especially since the authors claim that the "animals were reared in commercial
as well as backyard farms, were of different breed, age and sex, and were fed with food supplemented
with probiotics" (lines 137-138), while the references relate to a single sampling. Moreover, since they
use cultivated isolates, the choice of cultivation media may affect the outcome.

Response: Thank you very much for the comment and misunderstanding. All the information now can be
found in Table S1.

4. In lines 319-320, the authors claim that "only minority (31/6545) of genes suspected to HGT were
involved in the resistance mechanism". However, previous studies (e.g., references 17 and 26), including
the authors' previous studies (e.g., ref 19), indicate a much higher prevalence of antibiotic resistance
genes in the mobilome. Moreover, immediately after, in the paragraph starting in line 321, the authors
claim that their findings are consistent with previous studies and that the resistance genes were detected
and enriched. This paragraph is unclear, I believe the findings regarding the resistance genes should be
further discussed and clarified. If indeed the resistance genes are a minority, this is a very interesting

finding. Could it be attributed to the less strict criteria used which may lead to the detection of HGT
events prior to the extensive use of antibiotics?

Response: Thank you very much for the comment, you and another reviewer were right, the paragraph
was difficult to follow. Nevertheless, the abundance of ARG (< 0.5%) among identified HGT genes does
not say anything about the prevalence across the genera. To clarify our findings, we added some more
analysis and rewrite the paragraph – see lines 347-358.

5. In lines 83-88, the authors describe the advantages of using culture-independent techniques, but then
they use genome sequences of cultivated bacteria for their study. I think there should be an explanation
about the advantages of studying cultured systems.

Response: This is a little tough comment. Reviewer #1 ask us to add notes about experimental
approaches, Reviewer #2 to add notes about culture-dependent techniques, Reviewer #4 to remove notes
about culture-independent techniques and Reviewers #4 and #6 to focus on current bioinformatic
techniques to detect HGT. We modified the text in l. 84-110.

6. The authors emphasize the finding of genes with unknown functions as one of the highlights of their
findings. However, such genes are very common, and according to Figure 5, they are actually
underrepresented in the co-shared genes compared to their abundance in the entire genomes of the
isolates.

Response: we checked our analysis regarding the genes of unknown function and also regarding COG
category S, genes of general function. And agreed, genes are common, and no enrichment or depletion
have been detected. Therefore have we deleted inappropriate findings and statements throughout the ms,
especially in the Abstract and Conclusion sections.

7. In Figure 1, panel III] it says "identification of nearly identical protein sequences" while the nucleic
acids sequences (and not proteins sequences) were used in the pipeline. Please correct.

Response: done

8. In Figure 6, the authors should add a sentence to the figure legend explaining what the nodes and edges
represent.

Response: done.

9. In line 224 it says "888 genes were co-shared across 8 phyla", whereas Table 1 shows that there are 926
co-shared genes across 8 phyla (and 888 of them are known genes).

Response: Thanks for the comment. Table 1 was corrected.

10. The paragraph starting in line 321 is not clear.

Response: It has been already discussed in point 2.

11. In general, I feel that the manuscript would benefit from professional editing, as some parts of it are
not coherent and are hard to understand.

Response: Thank you for the comment. The manuscript has been reviewed by layperson in terms of
genomics as well as by native speaker.

Reviewer #4 (Comments for the Author):

In this manuscript, the authors describe the use of a bioinformatic analysis pipeline aiming to identify
horizontally transferred genes among a collection of genome sequences. The identification of HGT-genes
is based on: i) identification of shared genes with high identity and coverage among all genomes,
genomospecies, and within higher order taxa, ii) statistical comparison of COG profiles present in the
shared gene pool of a determined taxonomic level VS the "all" gene pool, and iii) analysis of the highly
similar genes shared by taxa showing an statistically different COG profile from the "All" COG profile.
They apply this pipeline on 452 draft genomes obtained from bacterial isolates that belong to pig and
chicken intestinal microbiota.

Response: Thank you for dedicating time to consider and review our manuscript. We highly appreciate the
feedback provided by Reviewer #4, which we have diligently incorporated into the revised manuscript.

Major comments:

1. It is not clear whether the manuscript is oriented to present the authors' findings about HGT in the pig
and chicken microbiota or the utility of their pipeline. The title seems to indicate the second option.
However, throughout the manuscript, it is not clear how the applied pipeline compares to other previously
published pipelines/software in terms of advantages, limitations, and major applications (Which identity
thresholds should be applied? How different identity thresholds affect the results? Are there HGT-genes
that are overlooked by the proposed pipeline? Which were those genes?). On the other hand, the finding
that nearly identical genes shared by different bacterial genera likely participated in HGT is not a new
observation. Please, include a discussion about your findings in terms of the antimicrobial-resistance
genes found.

Response: Thanks you very much for the comment. The focus of the manuscript is to introduce an
innovative computational tool for detecting HGT genes within genome or metagenome collections. We
have extensively revised a significant portion of the manuscript to address this topic convincingly with
the reviewers' comments.

2. What are the findings if the genes shared at the Family level are analyzed? Dos the COG categories
change o remain the same? IS there an enrichment of a particular COG category likely involved in HGT?
Response: we included information about Family and Phylum levels, lines 319-322.

3. The Methods Section needs to be rewritten to be more friendly with the reader in order to provide a
guide to follow the proposed pipeline. See an example here:
<https://www.frontiersin.org/articles/10.3389/fenvs.2022.901917/full>.

Response: thank you for the comment. We have carefully reviewed the section once again and have
worked on improving its clarity.

4. Is it possible to provide the command line used by the authors to apply this pipeline? This would
provide to be highly helpful for the researchers trying to apply this pipeline.

Response: We have provided the parameter settings used to run the pipelines in Supplementary File 1.

5. In my opinion, contrary to what is stated in the Conclusions, this pipeline is not as intuitive and has
many limitations. However, it seems to be very helpful to provide a preliminary set of shared genes that
can be further characterized or analyzed in order to assess their involvement in HGT. I believe the
conclusions should be centered on this.

Response we slightly modified the text in the Conclusion and avoid using the word “intuitive” throughout
the manuscript.

6. While the title states that the pipeline identify mobilome activity, the results only show the
identification of highly identical shared genes annotated as encoding proteins involved in mobilization. In
my opinion this cannot be stated as detection of "mobilome activity".

Response: Thank you for your suggestion, we mostly replaced “mobilome activity” with “involved in
mobility” such as in the title. Regarding the Title itself, Reviewer #1, #2, and #4 also asked us to make
some changes, therefore the manuscript is entitled “Detecting Horizontal Gene Transfer among
Microbiota: An Innovative Pipeline for Identifying Co-Shared Genes within the Mobilome through
Advanced Comparative Analysis.”

7. The introduction needs to be restructured to better address the importance of the detection of HGT in
the animal gut microbiota, the current detection pipelines/approaches and the necessity of a pipeline that
can do what the authors propose.

Response: The Introduction was modified and enlarged (l. 74-114).

8. Lines 103-112. Is there missing text here? What were those 8 phyla? Additionally, the findings
described in lines 108-112 are not shown in the phylogenetic trees provided, which are colored only by
Family.

Response: The name of phyla were added to the ms (126-128). Regarding the phylogenetic trees (Figure
1, S1), the corresponding legends have been slightly modified: *Bacteroidetes* are purple, *Firmicutes*
green, *Proteobacteria* blue, *Actinobacteria* yellow, *Fusobacteria* orange, *Verrucomicrobia* beige,
*Elusimicrobia* golden brown, *Synergistetes* pink.

9. Figures require adjustments for a more detailed presentation.

a. Fig 2. Require the identification of the bacterial phyla. Brackets or an additional color ring (since iTOL
was used) could be added.

Response: Regarding the phylogenetic trees (now Figure 1, S1), the corresponding legends have been
slightly modified: *Bacteroidetes* are purple, *Firmicutes* green, *Proteobacteria* blue, *Actinobacteria* yellow,
*Fusobacteria* orange, *Verrucomicrobia* beige, *Elusimicrobia* golden brown, *Synergistetes* pink.

b. Fig. 3. Please modify the figure to better convey the information. How were the genomospecies
grouped in the 7 phyla? How this "sub-grouping improved and fastened the comparison"?

Response: We modified the legend as folloes: . An unrooted phylogenetic tree generated through UBCG
analysis was employed to categorize the genome sequences into 7 distinct phylogenetically distant
groups: Lactobacillales (Firmicutes, group1), Lachnospirales (Firmicutes, group2), Negativicutes
(Firmicutes, group3), Oscillospirales (Firmicutes, group4), Bacteroidetes (group 5), Proteobacteria (group
6) and Actionobactera (group7). Within each of these groups, the genome sequences were subjected to
comparison using dRep. The objective of this method was to define clusters of genomes categorized as
the same genomospecies, identified as a group of isolates with an average nucleotide identity (ANI) of \geq
95%. The results of dRep clustering were also reconciled during the identification of more distant related
taxa, such as genera and families. Without subgrouping, the analysis of Firmicutes failed due to
polyphyletic nature of the phylum. In addition, sub-grouping also enhanced and expedited the
comparison.

c. Fig.4. Shared genes between the same or closely related species dominate the heatmap. The genes
shared by more distantly related species are difficult to see. Please, correct this. You could use a different
color instead of white in the "number of genes" scale that allows you to highlight the shared genes that are
currently showed in very light blue color.

Response: we added the Figure S3 to improve visibility

310 d. Fig. 5. Please clearly indicate what are the comparisons (which group VS which group, or the subject
of the comparison) that resulted in a $p < 0.05$. You could use brackets or lines. Figure 5 could be used to
explain how the comparison of COG profiles results in identification of the HGT-genes since this is not
clear in the main text.

Response: done

e. Fig. 6. Each color represents a Family, however there are many similar colors, difficulting the visual
identification. Please, write the names of important families or Families involved in important findings,
near their location in the corresponding networks.

Response: done

10. In Table 1. Is not clear whether the Identified genes suspicious of HGT are from the non-redundant
pool (NRPG) or from the 1,235,343 pool. Also, it would be helpful to provide the numbers of the NRPG
genes identified. Also, are the "unknown genes" genes of unknown function? Please clarify or correct.

Response: we incorporated the information about number of genomospecies in Table 1. Clarified text can
be seen in lines 191-193.

11. Lines 219-221. This explanation of 16S identity thresholds belong to the Methods section.

Response: we agree with you, however we kept the sentence in the text to understand the following text
more easily.

12. Methods Section, line 344. The procedence of the bacterial culture collection is unclear. Please detail
(or cite and summarize) the procedure of isolation of the bacterial strains. Also, it is unclear if the
additional 173 genomes are part of the 452 or not. Please clarify.

Response: Thank you very much for the comment and misunderstanding. All the information now can be
found in Table S1.

13. Line 279. "We consider co-shared genes between different genera are very likely to be mobilized." It
is not clear how the previously presented reasoning supports this conclusion. Please, clarify.

Response: We do agree that this is a little misleading information which has no reason to be stated here.
The information that genes shared by different genera are likely to be connect with HGT has already been
discussed in the "Identification and statistical verification of HGT gene pools" section, specifically lines
282-283.

14. Methods Section, lines 418-422. It is not clear what is the Dunn's test comparing. Also in Fig. 5. Is it a
given COG profile VS the "All" category? Or VS any other category? Moreover, the text does not explain
how comparison of COG categories results in identification of HGT genes which is the main focus of the
paper. Based on what is written in the Results/Discussion section in Line 239-244, it seems to be a
comparison between the COG profile of a given group VS the "All" group. However, this is not
understood from Fig. 5 and is not described in the Methods Section.

Response: To clarify our hypothesis, we added more description in the legend of Figure 5 and in the
Method section (lines 533-537).

15. Related to the previous comment. How is the statistical analysis carried out? I'm not sure how a
categorical COG profile can be compared to another with the Friedman and Dunn tests?

Response: Friedman's test (a non-parametric statistical test) was first utilized to assess whether there were
statistically significant differences in COG profiles (using either absolute or relative frequencies) among
different gene sets. The comparison of COG profiles using absolute frequencies was statistically
significant. Therefore, Dunn's post hoc test was employed to assess the specific groups that exhibited
differences The COG profile of each identified set of HGT genes was compared to the "All" group.

Minor comments:
Line 34: Is it necessary to write "co-shared"? "Shared" already conveys the idea of all the involved
subjects (genomes) having something (gene) in common.

Response: That's a good point. To us, the term "co-shared" accentuates the statement, particularly when
the genes are shared by two entities (e.g., genera).

Line 46: "only several genes were co-shared between." Please rephrase to specify how many genes.

Response: done.

Lines 66-68: Does HGT is the consequence or the cause of the adaptation/competition capacity of
microorganisms? The cited reference does not deal with HGT. Please cite the correct reference supporting
the claim.

Response: you are right, we improve the text as follows: HGT-mediated gene gain and loss is often the
consequence of adaptation to environmental changes under strong selection pressure such as in the case of
multi-drug resistant bacteria (2). On the other hand, HGT can also be a cause of adaptation: by acquiring
genes from other organisms, microorganisms can rapidly gain new functions, traits, or metabolic
pathways that improve their ability to compete with other microorganisms occurring in the same
environment.

The reference was modified.

Line 77: How does the conventional methods for HGT detection rely on .. the analysis among multidrug
resistant pathogens? The meaning of this sentence is unclear.

Response: we clarified the sentence (l. 74-76) as follows: The conventional methods of HGT detection,
particularly in clinical settings, relied on the comparative genomic analysis of closely related taxa (9) or
the analysis of complete genome sequences of multidrug-resistant pathogens, especially of
Enterobacteriaceae family (10).

Hope, our ideas are now more clear.

Lines 78-80: the phrase in these lines seems to be more related to the previous paragraph than the
paragraph in which is currently placed.

Response: you are right. We re-structuralized and modified this part of the Introduction (l. 76-84).

Line 84: Here the utility of culture-independent techniques is presented. However, below it is stated that
the work was carried out with isolated bacteria. Please remove the phrase about culture-independent
techniques.

Response: This is a little tough comment. Reviewer #1 ask us to add notes about experimental
approaches, Reviewer #2 to add notes about culture-dependent techniques, Reviewer #4 to remove notes
about culture-independent techniques and Reviewers #4 and #6 to focus on current bioinformatic
techniques to detect HGT. We modified the text in l. 84-110.

Line 94: "genes suspicious suspected of for HGT". Please correct.

Response. We corrected it as follows: genes likely suspicious for the HGT (.114-115)

Line 117: Please reference the corresponding figure showing what was described in the text.
Response: done.

Line 120: "To identify HGT genes" instead of "traits".
Response: done

Line 131: Please, indicate the statistical analysis used.
Response: done

Line 145: Please, correct the tense of the text, since the past tense here gives the notion that you are
describing your findings. However, for the cited reference, you are actually presenting a previous
published observation.
Response: done

Line 150: Please, reference the corresponding figure showing what is described in the text.
Response: done

Line 265: The sentence "In total, 6,545 unique genes were co-shared by at least two genera" can be
understood as there are two genera that share 6545 genes. Is this correct? Please, clarify or correct.
Response: I modified the statement as follows: In total, 6,545 unique genes were co-shared across
different genera. Hope, this is more clear now.

Line 377: how does the evolutionary relationship was estimated before using dRep?
Response: First, draft genome sequences were divided into 7 phylogenetically distant groups (Figure 3)
based on the unrooted phylogenetic tree generated through UBCG analysis (Figure 1). Within each of
these groups, the genome sequences were subjected to comparison using dRep.

Line 388: why a 300 bp threshold was selected? Please, indicate the rationale and include a short
discussion (in the discussion section) of how the threshold affects the results. For example, some small
genes involved in HGT, such as recombination directionality factors, have ≈ 200 bp in length.
Response: This is a really good and important comment. We included rationale for the threshold in lines
154-162, and made some comment in lines 390-394.

Lines 406-408. Please, specify how the comparison of COG profiles was used to identify HGT genes.

Response: we revised the text to improve clarity, see lines 533-537.

Line 431. "No chickens have been sacrificed solely for the purpose of this study." Does the "solely" word
means that chickens were in fact euthanized but not only for this research project? Please, clarify. Also,
assess whether "euthanized" is a more proper word instead of "sacrificed".

Response: No chickens or pigs have been euthanized for the purpose of this study.

Reviewer #6 (Comments for the Author):

The manuscript by Schwarzerova et al. introduces a potentially valuable bioinformatic pipeline for
detecting laterally transferred genes (LGT) between bacteria. The authors claim that by searching for
nearly identical gene sequences in phylogenetically unrelated genomes, they can identify recent cases of
LGT and shed light on the emergence of antibiotic resistance. While the research topic is promising, the
current version of the manuscript falls short of conveying the true value of the data and requires further
attention from the authors.

Response: Thank you for dedicating time to consider and review our manuscript. We highly appreciate the
feedback provided by Reviewer #6, which we have diligently incorporated into the revised manuscript.

To strengthen their work, the authors should compare their novel method with existing approaches and
analyze a new dataset. It is crucial for the authors to demonstrate the advantages and limitations of their
method in comparison to other techniques and showcase what can be uncovered using their pipeline. They
could start by applying several existing LGT detection methods to a well-analyzed set of bacterial
genomes, followed by implementing their own method. Each identified case should undergo verification
through multiple independent statistical techniques, and the significance measures should be compared.
Only after this thorough analysis, the authors should introduce their dataset derived from chickens and
pigs.

Response: We wholeheartedly agree with this comment and have consequently incorporated this type of
analysis into the manuscript. We added short description of the current detection pipelines to the
Introduction (l. 93-110), Results and Discussion (l. 360-441), and Method (l. 547-553) sections and Table
5 was provided.

Regarding the pipeline itself, the verification of the detected cases lacks persuasiveness. While the authors
employed COG groupings, additional tests are required to ensure the reliability of the results.

Response: Agreed. To clarify our hypothesis, we added more description in the legend of Figure 5 and in
the Method section (lines 533-537). Although the statistic analysis can be improved, other authors
confirmed our findings (line 279).

Furthermore, the manuscript lacks information on how the authors assigned taxonomic names to the 400+
isolates. It is essential for the authors to clarify this aspect. Additionally, reference 33 presents a smaller
set of isolates, which raises questions about the identification of bacteria used in the study.

Response: Thank you for the comment. Taxonomical assignments and nomenclature of each genome were
determined using the BLASTn comparison against NCBI RefSeq16S rRNA sequence database, as of
March 31 2023 (lines 459-460). Alternatively we also provided current nomenclature according the
current GTDB (lines 471-472), see Table S1

A significant drawback of the manuscript is the combined presentation of Results and Discussions. This
approach hinders the reader's ability to extract the true value of the work. It can be acceptable for some
descriptive studies but not for a bioinformatic pipeline. The authors should clearly distinguish between
these sections and provide a concise summary of their findings. As it stands, it is unclear whether the
manuscript primarily focuses on the development of a novel pipeline or the analysis of the mobilomes of
gut bacteria. Neither of these topics is adequately described, leading to an overall lack of clarity and
limited incremental value of the manuscript.

Response: Thank you for the comment. Thanks you very much for the comment. The focus of the
manuscript is to introduce an innovative computational tool for detecting HGT genes within genome or
metagenome collections. We have extensively revised a significant portion of the manuscript to address
this topic convincingly with the reviewers' comments including Discussion section.

*Minor: the ms text needs revision regarding overstatements and the use of terms*

Response: Agreed, we slightly lower the overstatements and add limitation of the purposed pipeline.
Regarding the use of jargon/terms, I have found the following terms within the manuscript: NCBI, HGT,
MGE, IS, ARG, COG. To my opinion, all are commonly used in genomics and horizontal gene transfer-
related manuscripts. Other term (e.g. GTDB, dRep, MetaCHIP) are directly the names of the pipeline or
database, no abbreviation has been introduced by us. And lastly, we used term NRPG (non-redundant pan-
genome) throughout the text. However, I do understand that for novice and layman in the subject, it is
difficult to understand all terms.

September 22, 2023

Dr. Darina Cejkova
Vysoke uceni technicke v Brne
Department of Biomedical Engineering
Technicka 12
Brno 61600
Czech Republic

Re: Spectrum01964-23R1 (Detecting Horizontal Gene Transfer among Microbiota: An Innovative Pipeline for Identifying Co-Shared Genes within the Mobilome through Advanced Comparative Analysis)

Dear Dr. Darina Cejkova:

Thank you for submitting your manuscript to Microbiology Spectrum. As you will see your paper is very close to acceptance. Please modify the manuscript along the lines I have recommended. As these revisions are quite minor, I expect that you should be able to turn in the revised paper in less than 30 days, if not sooner. If your manuscript was reviewed, you will find the reviewers' comments below.

When submitting the revised version of your paper, please provide (1) point-by-point responses to the issues raised by the reviewers as file type "Response to Reviewers," not in your cover letter, and (2) a PDF file that indicates the changes from the original submission (by highlighting or underlining the changes) as file type "Marked Up Manuscript - For Review Only". Please use this link to submit your revised manuscript. Detailed instructions on submitting your revised paper are below.

Link Not Available

Sincerely,

Feng Gao

Reviewer comments:

Reviewer #12 (Comments for the Author):

The authors have sufficiently addressed reviewer feedback.

Reviewer #2 (Comments for the Author):

The authors have submitted a revised version of this manuscript along with a rebuttal letter addressing the reviewers' comments on the original manuscript. The revised manuscript underwent significant revisions, successfully addressing all the concerns I raised in my initial review. Most important, the overall readability of the paper was significantly improved.

Few minor comments:

1. Authors replaced the term "novel" with the term "innovative". I still don't like the use of the new term as well, but I can live with that.
2. Lines 33-34: "we propose that nearly identical genes co-shared between distinct genera can be mobilized from genome to genome via HGT". I assume authors mean that nearly identical genes cos-shared between distinct genera can be evidence for a previous event of mobilization of that gene from genome to genome via HGT.

3. Line 96: missing space before ref (18).
4. Line 184: "Then, genomospecies definition according tog were in concordance". Not clear. What "tog" means?
5. Lines 189-190: "Only one such gene was retained the others were discarded from the (species) pan-genome." Don't you miss the word "while" between two parts of the sentence?
6. Line 322: shouldn't have a comma before (Q).
7. Line 370: two commas. One should be removed.
8. Line 402: "Nevertheless we ale performed". Do you mean "Nevertheless, we also performed"?
9. Lines 409-410: "we detected 74 vs. 77 HGT genes (Table 5) while 43 identical alleles were identified as 30 nearly identical alleles discovered as by the 100id pipeline and the proposed pipeline." Not clear.

Reviewer #4 (Comments for the Author):

Reviewer #4

I would like to thank the authors for their detailed responses, especially when receiving comments from so many reviewers. Almost all my comments have been successfully addressed, and I would like to underscore the followings that I consider need a little more work.

(numbers correspond to the original comment)

8. Lines 103-112. Is there missing text here? What were those 8 phyla? Additionally, the findings described in lines 108-112 are not shown in the phylogenetic trees provided, which are colored only by Family.
9. Figures require adjustments for a more detailed presentation.
 - a. Fig 2. Require the identification of the bacterial phyla. Brackets or an additional color ring (since iTOL was used) could be added.

Regarding comments 8 and 9. Figure 1 still requires more adjustment. Name of families are highly difficult to see in the tree and this information is redundant with the legend that also indicate the bacterial families. Please, also modify title of the legend to indicate what is showing. While using colors to identify phyla, the different shades of the same color are confusing. Please add another iTOL colored strip to indicate phyla unambiguously (<https://itol.embl.de/help.cgi#strip>). The size proportion between legend and the phylogenetic tree also needs to be adjusted. The legend is too large compared to the tree.

10. In Table 1. Is not clear whether the Identified genes suspicious of HGT are from the non-redundant 322 pool (NRPG) or from the 1,235,343 pool. Also, it would be helpful to provide the numbers of the NRPG 323 genes identified. Also, are the "unknown genes" genes of unknown function? Please clarify or correct.

The corrected table is fine. However, in the corrected manuscript is stated that "Eighteen genomospecies did not co-share any gene at all (Figure 4)". Do these 18 genomospecies need to be subtracted in Table 1 column Genomospecies under "Genes suspicious of HGT"? If that is the case, please clearly indicate that the 10,629 suspicious genes were found among 231 out of 249 genomospecies. Additionally, the 18 genomospecies that did not shared genes can not be seen in Figure 4. Please indicate de appropriate supplemental file where this information is.

Preparing Revision Guidelines

Please return the manuscript within 60 days; if you cannot complete the modification within this time period, please contact me. If you do not wish to modify the manuscript and prefer to submit it to another journal, please notify me of your decision immediately so that the manuscript may be formally withdrawn from consideration by Microbiology Spectrum.

Review for Spectrum01964-23_R1: “Detecting Horizontal Gene Transfer among Microbiota: An Innovative Pipeline for Identifying Co-Shared Genes within the Mobilome through Advanced Comparative Analysis”.

The authors have submitted a revised version of this manuscript along with a rebuttal letter addressing the reviewers' comments on the original manuscript. The revised manuscript underwent significant revisions, successfully addressing all the concerns I raised in my initial review. Most important, the overall readability of the paper was significantly improved.

Few minor comments:

1. Authors replaced the term “novel” with the term “innovative”. I still don't like the use of the new term as well, but I can live with that.
2. Lines 33-34: “we propose that nearly identical genes co-shared between distinct genera can be mobilized from genome to genome via HGT”. I assume authors mean that nearly identical genes cos-shared between distinct genera can be evidence for a previous event of mobilization of that gene from genome to genome via HGT.
3. Line 96: missing space before ref (18).
4. Line 184: “Then, genomospecies definition according tog were in concordance”. Not clear. What “tog” means?
5. Lines 189-190: “Only one such gene was retained the others were discarded from the (species) pan-genome.” Don't you miss the word “while” between two parts of the sentence?
6. Line 322: shouldn't have a comma before (Q).
7. Line 370: two commas. One should be removed.
8. Line 402: “Nevertheless we ale performed”. Do you mean “Nevertheless, we also performed”?
9. Lines 409-410: “we detected 74 vs. 77 HGT genes (Table 5) while 43 identical alleles were identified as 30 nearly identical alleles discovered as by the 100id pipeline and the proposed pipeline.” Not clear.

The authors have sufficiently addressed reviewer feedback.

Dear editor Feng Gao,
You will find all the modifications made to the revised manuscript highlighted in yellow within the "Marked Up Manuscript" file. We also provided responses to additional comments below in this letter.

Reviewer #2 (Comments for the Author):

The authors have submitted a revised version of this manuscript along with a rebuttal letter addressing the reviewers' comments on the original manuscript. The revised manuscript underwent significant revisions, successfully addressing all the concerns I raised in my initial review. Most important, the overall readability of the paper was significantly improved.

Few minor comments:

1. Authors replaced the term "novel" with the term "innovative". I still don't like the use of the new term as well, but I can live with that.

You are welcome.

2. Lines 33-34: "we propose that nearly identical genes co-shared between distinct genera can be mobilized from genome to genome via HGT". I assume authors mean that nearly identical genes co-shared between distinct genera can be evidence for a previous event of mobilization of that gene from genome to genome via HGT.

Thank you very much for the comment to improve readability of the ms. We changed the sentence according to your suggestion.

3. Line 96: missing space before ref (18).

Done

4. Line 184: "Then, genomospecies definition according tog were in concordance". Not clear. What "tog" means?

I changed the sentence as follows: "Finally, in defining genomospecies using the UBCG tree, 16S rDNA, and dRep clustering, they exhibited concordance. Additionally, the GTDB taxonomy mostly aligned with these results".

5. Lines 189-190: "Only one such gene was retained the others were discarded from the (species) pan-genome." Don't you miss the word "while" between two parts of the sentence?

Agreed, thank you for notifying us.

6. Line 322: shouldn't have a comma before (Q).

Agreed, thank you for notifying us.

7. Line 370: two commas. One should be removed.

Agreed, thank you for notifying us.

8. Line 402: “Nevertheless we ale performed”. Do you mean “Nevertheless, we also performed”?

Agreed, thank you for notifying us.

9. Lines 409-410: “we detected 74 vs. 77 HGT genes (Table 5) while 43 identical alleles were identified
as 30 nearly identical alleles discovered as by the 100id pipeline and the proposed pipeline.” Not clear.

I agree that the detailed results presented may appear controversial and awkward, but they align with
mathematical definitions. On a global scale, employing a 100% identity (100id) versus a 99% identity
(99id) pipeline, we detected 74 and 77 HGT genes identified by at least two different computational tools
mentioned in the manuscript, respectively. The differentiation between 30 and 43 HGT genes stems from
a direct comparison of the 100id versus 99id pipelines, aiming to identify HGT genes shared across
various genera. It is important to note that the number of gene pools shared by distinct 99% identity genes
(or allele variants) increased when we categorized them into different 100% identity gene variants.
Simply put, some of the 99% genes were split into multiple 100id gene variants. Conversely, not all 100id
variants were co-shared by different genera. Therefore, precise number of HGT genes cannot be easily
predicted when shifting the parameters of the pipeline, and in our case it dropped from the number 43 to
30.

In Table 5, we aimed to compare the overall impact of the proposed pipeline, incorporating your
suggestion of 99id and 100id gene variants, on the detection of the HGT gene pool in comparison to other
computational methods. Generally, the results demonstrated a similarity in the number of HGT genes (74
vs. 77); however, interpreting the details requires a more nuanced approach. For this reason, within the
body of the ms (l. 412), we have outlined only major impacts of the comparison.

Reviewer #12 (Comments for the Author):

The authors have sufficiently addressed reviewer feedback.

Thank you for the help to improve our manuscript.

Reviewer #4 (Comments for the Author):

I would like to thank the authors for their detailed responses, especially when receiving comments from
so many reviewers. Almost all my comments have been successfully addressed, and I would like to
underscore the followings that I consider need a little more work.

(numbers correspond to the original comment)

8. Lines 103-112. Is there missing text here? What were those 8 phyla? Additionally, the findings
described in lines 108-112 are not shown in the phylogenetic trees provided, which are colored only by
Family.

9. Figures require adjustments for a more detailed presentation.

a. Fig 2. Require the identification of the bacterial phyla. Brackets or an additional color ring (since iTOL
was used) could be added.

Regarding comments 8 and 9. **Figure 1 still requires more adjustment. Name of families are highly**
**difficult to see in the tree and this information is redundant with the legend that also indicate the**
**bacterial families. Please, also modify title of the legend to indicate what is showing. While using**
**colors to identify phyla, the different shades of the same color are confusing. Please add another**
**iTOL colored strip to indicate phyla unambiguously (<https://itol.embl.de/help.cgi#strip>). The size**
**proportion between legend and the phylogenetic tree also needs to be adjusted. The legend is too**
**large compared to the tree.**

The figures and legends were altered based on your suggestions. We can agree that visual and written
captions convey the same information. However, visual identification is more comprehensible at first
glance, while detailed written descriptions allow for precise identification of specific families (phyla),
especially when considering potential inaccuracies in taxonomic identification.

10. In Table 1. Is not clear whether the Identified genes suspicious of HGT are from the non-redundant
322 pool (NRPG) or from the 1,235,343 pool. Also, it would be helpful to provide the numbers of the
NRPG 323 genes identified. Also, are the "unknown genes" genes of unknown function? Please clarify or
correct.

**The corrected table is fine. However, in the corrected manuscript is stated that "Eighteen**
**genomospecies did not co-share any gene at all (Figure 4)". Do these 18 genomospecies need to be**
**subtracted in Table 1 column Genomospecies under "Genes suspicious of HGT"? If that is the case,**
**please clearly indicate that the 10,629 suspicious genes were found among 231 out of 249**
**genomospecies. Additionally, the 18 genomospecies that did not shared genes can not be seen in**
**Figure 4. Please indicate de appropriate supplemental file where this information is.**

Thank you for your feedback. We have addressed your concern regarding the informativeness of the
statement. In Table 1, we have modified the description to explicitly state the total number of examined
taxa for clarity. Additionally, we have made revisions to lines 193-196 to enhance the overall
informativeness of the text.

For a more detailed understanding of gene transfer dynamics, we have provided detailed information in
Figure 4 and Figure S2. Should you desire further comprehensive information we can provide it
personally, but this information is beyond the scope of the ms.

Re: Spectrum01964-23R2 (Detecting Horizontal Gene Transfer among Microbiota: An Innovative Pipeline for Identifying Co-Shared Genes within the Mobilome through Advanced Comparative Analysis)

Dear Dr. Darina Cejkova:

Your manuscript has been accepted, and I am forwarding it to the ASM production staff for publication. Your paper will first be checked to make sure all elements meet the technical requirements. ASM staff will contact you if anything needs to be revised before copyediting and production can begin. Otherwise, you will be notified when your proofs are ready to be viewed.

Sincerely,
Feng Gao
Editor
Microbiology Spectrum